# Rational Transductors

**Mehryar Mohri** [1] [2]

## Abstract

Standard Transformers excel at semantic modeling but struggle with rigid sequential logic and state tracking. Theoretical work establishes that self-attention is limited to $\mathsf{AC}^0$ (under hard attention) or $\mathsf{TC}^0$ (under soft attention), complexity classes that often fail to support robust length generalization on sequential problems without intermediate chain-of-thought (Hahn, 2020; Merrill et al., 2022). In this work, we introduce *Rational Transductors*, a dual-stream architecture that augments the Transformer with a matrix-valued recurrence derived from Weighted Finite Automata (WFA). By injecting rational state information into the attention mechanism via a *Deep Rational Injection* scheme, our framework strictly generalizes Transformers to capture all Regular Languages, $\mathsf{NC}^1$-complete problems (such as Boolean Formula Evaluation), and fundamental separations like Parity and Modular Counting, while preserving $O(\log T)$ parallel training efficiency. Theoretical analysis and empirical results demonstrate that Rational Transductors solve the "Regular Gap," enabling robust length generalization on algorithmic tasks where standard Transformers fail, without the sequential computational bottlenecks of traditional RNNs.

## 1. Introduction

The Transformer architecture (Vaswani et al., 2017) has revolutionized sequence modeling, establishing itself as the de facto standard for natural language processing, code generation, and beyond (Brown et al., 2020; Touvron et al., 2023). Its success is largely attributed to the self-attention mechanism (Schmidhuber, 1992; Graves, 2013; Bahdanau et al., 2014; Luong et al., 2015), which models long-range semantic dependencies by allowing every token to interact directly with every other token. However, this semantic power comes with a well-documented blind spot: standard Transformers struggle with rigid *sequential logic* and *state tracking* (Liu et al., 2023; Bhattamishra et al., 2020). Theoretical analyses have shown that self-attention, without intermediate recurrence or chain-of-thought, is limited to $\mathsf{AC}^0$ (under hard attention) (Hahn, 2020) or $\mathsf{TC}^0$ (under soft attention) (Merrill & Sabharwal, 2023; Chiang, 2024), complexity classes that struggle to represent unbounded sequential dependencies uniformly (Huang et al., 2025; Merrill et al., 2022). While $\mathsf{TC}^0$ models can theoretically approximate tasks like parity, they lack the inductive bias to *learn* state-tracking solutions that generalize to unseen lengths. Specifically, standard Transformers often fail to learn robust solutions for tasks outside the C-RASP fragment (Yang et al., 2025a; Huang et al., 2025), such as modular counting. In practice, this manifests as brittleness: models trained on short contexts frequently fail to maintain consistent state (e.g., tracking variable values or nested brackets) when deployed on longer sequences (Anil et al., 2022).

To address these limitations, a resurgence of interest in Recurrent Neural Networks (RNNs) and State Space Models (SSMs) has emerged (Gu et al., 2022; Gu & Dao, 2023; Smith et al., 2022). These architectures reintroduce a latent state that evolves over time, theoretically enabling infinite context tracking. However, they often face a structural trade-off: simple time-invariant recurrences (like S4) are efficient but lack expressivity. While recent selective state space models (like Mamba) introduce token-dependence to bridge this gap, they typically do so by interleaving recurrence into the deep backbone, which reintroduces layer-wise sequential dependencies during inference or training. In contrast, simple gated RNNs (like LSTM) remain fundamentally sequential.

In this work, we argue that the dichotomy between "Attention" (semantics) and "Recurrence" (syntax) is a false one. We introduce *Rational Transductors*, a dual-stream architecture that unifies the semantic flexibility of Transformers with the rigorous state-tracking capabilities of Weighted Finite Automata (WFA). Our approach is grounded in the formal theory of Rational Power Series (Schützenberger, 1961), which provides the mathematical foundation for regular languages and their quantitative generalizations.

We argue that the failure of Transformers to generalize on algorithmic tasks is not due to a lack of capacity, but a lack

---

[1]Google Research, New York, NY; [2]Courant Institute of Mathematical Sciences, New York, NY. Correspondence to: Mehryar Mohri <mohri@google.com>.

*Proceedings of the $43^{rd}$ International Conference on Machine Learning*, Seoul, South Korea. PMLR 306, 2026. Copyright 2026 by the author(s).

of *syntactic inductive bias*. To correct this, we augment the Transformer with a *Rational Feature Head*, a matrix-valued recurrence that acts as a dedicated co-processor for sequential logic. Crucially, unlike standard RNNs that use non-linear activations ($\tanh$, sigmoid), our rational states evolve via linear matrix multiplication. This design choice yields two decisive advantages:

1. Parallel Scalability: The linear recurrence $\mathbf{h}_t = \mathsf{M}_{x_t} \mathbf{h}_{t-1}$ can be computed via parallel associative scans (prefix sums) in $O(\log T)$ time (Blelloch, 1990; Ladner & Fischer, 1980), bypassing the sequential bottleneck that plagues traditional RNNs.

2. Theoretical Transparency: The state dynamics correspond exactly to Weighted Finite Automata (Mohri, 2009; Droste et al., 2009), allowing us to leverage formal language theory (Schützenberger, 1961; Fliess, 1974) to prove guarantees on expressivity and generalization.

We ground this architecture in a rigorous learning theory. First, analyzing the *Random Rational Feature* limit, we prove that a sufficiently wide, randomly initialized rational head acts as a universal basis for sequential dependencies. But, our primary contribution is the *Differentiable Rational Feature* regime, where the transition matrices are learned end-to-end. We prove that this learned regime strictly expands the expressivity of Transformers to capture all Regular Languages and $\mathsf{NC}^1$-complete problems, while maintaining numerical stability through novel spectral parameterizations.

**Related Work** Our work lies at the intersection of three active research streams:

**State Space Models and Linear RNNs.** Recent advances in efficient sequence modeling, including S4 (Gu et al., 2022), Mamba (Gu & Dao, 2023), RWKV (Peng et al., 2023), DeltaNet (Schlag et al., 2021), and Kimi Linear (Kimi Team et al., 2025), rely on linearizing the state update to enable parallel training. While recent architectures like Mamba have popularized "Linear RNNs" as efficient layers grounded in signal processing (Gu et al., 2022), our framework provides a complementary automata-theoretic characterization. The Rational Transductor framework formalizes this class, providing the first rigorous automata-theoretic proofs that such linear recurrences are sufficient to solve the "Regular Gap" and capture $\mathsf{NC}^1$-complete reasoning. Furthermore, unlike "Deep SSMs" (e.g., Mamba, H3 (Fu et al., 2023)) which interleave recurrence and mixing layers, Rational Transductors adopt a "Sidecar" design. By keeping the recurrent state evolution strictly input-driven, we decouple the state tracking (WFA) from the feature mixing (Transformer). This guarantees exact $O(\log T)$ parallel training without the sequential layer-wise dependency inherent to stacked SSMs, or the iterative approximations required for non-linear RNNs. We note that recent work on structured sparse transition matrices (Terzić et al., 2025) pursues a complementary approach to state tracking in SSMs, while stack-augmented Transformers (DuSell & Chiang, 2024) explore pushdown extensions for context-free capabilities; our framework differs by providing automata-theoretic guarantees for regular languages via a sidecar linear recurrence. The interleaved vs. sidecar trade-off is further analyzed in Appendix B.2: interleaving yields higher per-parameter expressivity via non-linear cascades (Theorem 14), but reintroduces a sequential training bottleneck of depth $O(L \log T)$.

**Expressivity of Transformers.** Hahn (2020) and Merrill & Sabharwal (2023) established the $\mathsf{AC}^0$ and $\mathsf{TC}^0$ upper bounds for Transformers (depending on attention hardness), later strengthened by Chiang (2024), highlighting their inability to robustly model sequential state. Our work provides a constructive proof that augmenting attention with linear recurrence is sufficient to break this barrier and capture $\mathsf{NC}^1$.

**Spectral Learning of Automata.** We draw inspiration from spectral learning algorithms for WFAs (Balle & Mohri, 2015; 2012; Hsu et al., 2009), effectively embedding a spectral extraction mechanism directly into the deep learning optimization loop via gradient descent.

**Paper Organization.** The remainder of this paper is organized as follows. Section 2 formally defines the Rational Transductor architecture, detailing the WFA formalism, parallel feature layers, and the Deep Rational Injection method. Section 3 introduces the Unified Scaled Cayley Parameterization and discusses alternative structures like DPLR and the Universal Transductor. Section 4 provides a rigorous theoretical analysis of the model's expressivity, proving it solves the "Regular Gap" and characterizing its circuit complexity. Section 5 analyzes the learning theory of the model, establishing results on the universality of random features, optimization stability, and generalization bounds. Finally, Section 6 presents empirical validation on tasks probing the Regular Gap, length generalization, and efficiency.

## 2. The Rational Features Framework

We view state tracking through the lens of Weighted Finite Automata (WFA). Formally, a WFA over the field of real numbers $\mathbb{R}$ is defined as a tuple $\mathcal{A} = (\Sigma, d, \boldsymbol{\alpha}, \{\mathsf{M}_\sigma\}_{\sigma \in \Sigma})$, where $\Sigma$ is the finite alphabet, $d \in \mathbb{N}$ is the dimension of the state space, $\boldsymbol{\alpha} \in \mathbb{R}^d$ is the initial state vector, and $\mathsf{M}_\sigma \in \mathbb{R}^{d \times d}$ is the transition matrix associated with token $\sigma$. Given a sequence of input tokens $x = (x_1, \ldots, x_T)$, we are interested in the sequence of vectors $\mathbf{h}_t \in \mathbb{R}^d$ (hidden states) produced by the automaton, which is defined as follows:

$$\mathbf{h}_t = \mathsf{M}_{x_t} \mathbf{h}_{t-1}, \quad \text{with } \mathbf{h}_0 = \boldsymbol{\alpha}. \tag{1}$$

The $i$-th component $h_{t,i}$ represents the sum of the weights of all paths labeled with the prefix $x_{1:t}$ that end at state $i$,

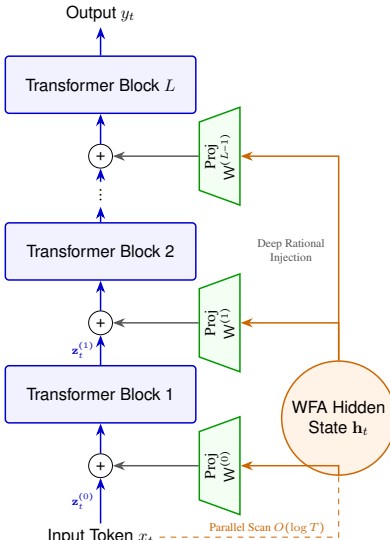

*Figure 1.* **The Rational Transducer Architecture.** The Rational Head (bottom right) extracts state variables $\mathbf{h}_t$ via a parallel scan. These states are injected into the Attention Stream via layer-specific projections $\mathsf{W}^{(l)}$, augmenting the semantic hidden states $\mathbf{z}_t^{(l)}$. This deep injection ensures state information is available at all depths without signal attenuation.

weighted by the initial values in $\boldsymbol{\alpha}$. We omit the final weight vector $\boldsymbol{\beta}$ in our definition as we are only interested in the sequence of intermediate vectors as features. We extend the definition of the matrices $\mathsf{M}_\sigma$ to sequences using the shorthand $\mathsf{M}_x = \mathsf{M}_{x_T} \cdots \mathsf{M}_{x_1}$, for $x = x_1 \ldots x_T$.

**Rational Feature Layers and Deep Integration.** We define the *rational feature vector* $\mathbf{h}_t$ at time step $t$ as the forward state of the automaton after processing the prefix $x_{1:t}$, that is $\mathbf{h}_t = \mathsf{M}_{x_{1:t}} \boldsymbol{\alpha}$. Unlike standard RNNs which use non-linear activation functions (e.g., $\tanh$ or $\sigma$), the update in (1) is linear. This linearity guarantees that the features capture Regular Languages (and their weighted generalizations) exactly (Schützenberger, 1961). Furthermore, the recurrence $\mathbf{h}_t = \mathsf{M}_{x_{1:t}} \boldsymbol{\alpha}$ can be computed efficiently on modern hardware using parallel associative scans (Blelloch, 1990). While the constant factor depends on matrix multiplication costs, we assume moderate $d$ (typically $d \in [4, 32]$) such that this overhead is negligible compared to the quadratic cost of attention.

A naive approach to integration would be to simply concatenate the rational feature vector $\mathbf{h}_t$ to the input token embedding. While standard residual connections theoretically allow information to propagate, relying on them forces the Transformer to preserve the exact state $\mathbf{h}_t$ within the semantic backbone $\mathbf{z}_t$, competing with feature extraction and suffering from signal attenuation due to repeated layer normalizations (Xiong et al., 2020). To address this, we propose *Deep Rational Injection* (see Figure 1).

Instead of augmenting only the input, we inject the rational

features directly into the hidden state of every Transformer block via an independent pathway. This ensures that a fresh, uncorrupted view of the precise state tracking information is available at all levels of abstraction. Let $\mathbf{z}_t^{(l)} \in \mathbb{R}^{d_{\text{model}}}$ denote the Transformer's hidden representation at time step $t$ immediately before layer $l$. We modify the input to each layer $l$ by adding a projected view of the rational state:

$$\widetilde{\mathbf{z}}_t^{(l)} = \mathbf{z}_t^{(l)} + \mathsf{W}_{\text{proj}}^{(l)} \mathbf{h}_t \tag{2}$$

where $\mathsf{W}_{\text{proj}}^{(l)} \in \mathbb{R}^{d_{\text{model}} \times d}$ is a learnable linear projection unique to layer $l$. By using a layer-specific projection $\mathsf{W}_{\text{proj}}^{(l)}$, the model can extract different aspects of the state history relevant to different depths of processing. For instance, early layers might use $\mathbf{h}_t$ for local syntactic parsing (e.g., matching parentheses), while deeper layers might use the same $\mathbf{h}_t$ to resolve long-term dependencies (e.g., tracking subject-verb agreement across long clauses). The modified stream $\widetilde{\mathbf{z}}_t^{(l)}$ is processed by the standard Self-Attention and Feed-Forward sub-layers. Crucially, because $\mathbf{h}_t$ is computed via a parallel scan, this deep injection adds no sequential dependency to the Transformer, preserving the $O(\log T)$ parallel training efficiency.

**Extended Architectural Analysis.** While our canonical architecture used a "sidecar" design with a single rational head, we explore hierarchical extensions in Appendix B. We analyze Stacked Rational Transducers, proving that while cascading linear automata does not strictly expand the function class (due to algebraic reducibility, Proposition 12), it significantly improves parameter efficiency for decomposable problems (Theorem 13). We further discuss non-linear stacking variants, which trade $O(\log T)$ parallel efficiency for increased expressivity via deep recurrence.

## 3. Parameterization

The Rational Transducer framework is agnostic to the specific internal structure of the transition matrices $\mathsf{M}_\sigma$. To solve the Regular Gap effectively, the transition matrices $\mathsf{M}_t$ must satisfy two conflicting requirements: (1) *Conservation:* They must support unitary dynamics ($\|\mathsf{M}_t\| = 1$) to preserve state over infinite horizons (e.g., for modular counting); (2) *Stability:* They must allow for contraction ($\|\mathsf{M}_t\| < 1$) to forget irrelevant history (e.g., for standard forgetting tasks) (Arjovsky et al., 2016).

### 3.1. Unified Scaled Cayley Parameterization

We propose a *Unified Scaled Cayley Parameterization* that captures both regimes in a single differentiable form. We define the transition matrix $\mathsf{M}_t$ as:

$$\mathsf{M}_t = g_t \cdot \mathcal{C}(\mathsf{A}_t) = g_t \cdot (\mathsf{I} + \mathsf{A}_t)(\mathsf{I} - \mathsf{A}_t)^{-1} \tag{3}$$

where $A_t \in \mathbb{R}^{d \times d}$ is a skew-symmetric matrix ($A_t^\top = -A_t$) and $g_t \in \mathbb{R}$ is a scalar gain factor. The parameters $A_t$ and $g_t$ are obtained via a linear projection of the input embedding $x_t$, allowing $M_t$ to be cached.

**Intrinsic Stability.** The Cayley transform $\mathcal{C}(A_t)$ maps the skew-symmetric algebra $\mathfrak{so}(d)$ to the special orthogonal group $SO(d)$ (Helgason, 2001). This structurally guarantees that the core transformation is a pure rotation with eigenvalues strictly on the unit circle ($\|\mathcal{C}\|_2 = 1$). Unlike unconstrained matrices, this parametrization prevents the "exploding/vanishing gradient" problem mechanistically, rather than relying on penalty terms (Pascanu et al., 2013). To ensure stable training, we adopt a *Near-Identity Initialization* scheme ($M_\sigma \approx I$) that biases the model toward long-term memory at the start (see Appendix E.3).

**The Scalar Component (Dynamics).** The scalar $g_t$ controls the energy of the recurrence: *Conservation Regime:* By fixing $g_t = 1$, the recurrence becomes strictly orthogonal ($\|M_t\|_2 = 1$). This is optimal for tasks like *Modulo Counting* or *Parity*, where state magnitude must be preserved indefinitely. *Decay Regime:* By learning $g_t = \sigma(\theta_t) \in (0, 1)$ (where $\theta_t$ is a learnable scalar), the model can emulate the gating mechanisms of LSTMs or SSMs, attenuating past information to focus on recent context.

### 3.2. Alternative Parameterizations

While our primary experiments focus on the Scaled Cayley form for its theoretical properties, the framework supports any structured matrix family.

**Diagonal Plus Low-Rank (DPLR):** $M_\sigma = D_\sigma + U_\sigma V_\sigma^\top$. This structure, popular in models like S4 and DeltaNet (Gu et al., 2022), allows for efficient $O(d)$ matrix-vector products. It is ideal for general sequence modeling where mixing and fading memory are required, but it lacks the exact unitarity needed for modulo counting.

**Shared Basis:** $M_\sigma = \sum_{k=1}^K a_{\sigma,k} B_k$. This parameterization reduces the number of parameters by sharing basis matrices $B_k$ across all tokens, inducing a low-rank tensor factorization (Yang & Hospedales, 2017). This acts as a strong inductive bias that stabilizes training when $|\Sigma|$ is large.

### 3.3. Parallel Combination and Direct Sums

We can construct a *Mixed Rational Head* by running multiple independent automata in parallel. Mathematically, this corresponds to the *Direct Sum* of the transition matrices:

$$M_\sigma = M_\sigma^{(1)} \oplus M_\sigma^{(2)} = \begin{pmatrix} M_\sigma^{(1)} & 0 \\ 0 & M_\sigma^{(2)} \end{pmatrix}. \quad (4)$$

This formulation allows the model to instantiate parallel heads with distinct dynamical biases (e.g., mixing a DPLR

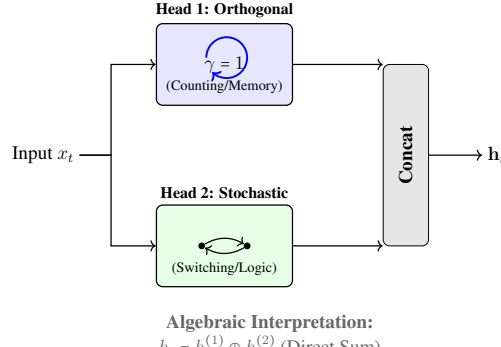

*Figure 2.* **The Universal Rational Transductor.** The architecture instantiates parallel heads with distinct dynamical biases: *Orthogonal* (top) for infinite memory and *Stochastic* (bottom) for discrete switching. These independent features are concatenated, corresponding to the direct sum ($\oplus$) of the underlying automata.

head for fading context with an Orthogonal head for exact counting). The resulting state dimension is additive ($d = d_1 + d_2$), maintaining efficiency. This principle enables the *Universal Rational Transductor*, where orthogonal and stochastic heads operate in parallel (Fig. 2).

**Remark on Topology and Reflections.** The Cayley transform maps to $SO(d)$ (determinant +1), which excludes reflections (determinant −1) necessary for tasks like Parity in minimal dimensions ($d = 2$). While one could augment Eq. (3) with a discrete sign flip mechanism, we find empirically that simply increasing the state dimension ($d \geq 3$) allows the model to embed reflections as rotations in a higher-dimensional space, resolving the topological obstruction without special casing.

## 4. Expressivity and Complexity

In this section, we rigorously analyze the expressive power of Rational Transductors. We demonstrate that the architecture bridges the "Regular Gap"—the fundamental inability of standard Transformers to uniformly represent periodic and sequential logic. While $\mathsf{AC}^0$ (hard attention) and $\mathsf{TC}^0$ (soft attention) Transformers struggle with tasks like modular counting without chain-of-thought (Hahn, 2020; Merrill et al., 2022), we show that Rational Transductors structurally capture these capabilities via linear recurrence.

We first establish that our framework strictly generalizes modern positional encodings. We then provide constructive proofs for key *Expressive Separations*, including Parity, Exact Modular Counting, and Exact Arithmetic Evaluation (Theorem 18), and conclude by situating the model within the algebraic hierarchy of automata via the Krohn-Rhodes theorem. Extended proofs and additional structural results—including the Expressive Hierarchy and Logical Completeness—are provided in Appendix C.

## 4.1. Positional Encodings as Rational Features

We first show that our framework strictly generalizes modern positional encoding schemes.

**Lemma 1** (Positional Encodings are Rational). *Standard Rotary Positional Embeddings (RoPE) (Su et al., 2021) are a special case of Rational Features where the transition matrices are input-independent unitary rotations.*

*Proof sketch.* Let $R_\theta$ be the block-diagonal rotation matrix from RoPE. Setting $M_\sigma = R_\theta$ for all $\sigma \in \Sigma$ yields the state evolution $h_t = R_\theta^t \alpha$, recovering absolute positional encodings exactly. Thus, Transductors inherently possess the capabilities of standard Transformers (via PE) while strictly extending them by enabling input-dependent transitions $M_\sigma$. $\square$

## 4.2. Solving the Regular Gap

**Theorem 2** (The Parity Gap). *Let $L_{parity}$ be the language of binary strings $x \in \{0,1\}^*$ containing an odd number of 1s. (1) A standard Transformer with fixed depth cannot uniformly recognize $L_{parity}$. (2) There exists a Rational Transductor with state dimension $d = 2$ that recognizes $L_{parity}$ with $100\%$ accuracy for any length $T$.*

*Proof.* Part 1 (Transformer Limitation): Theoretical lower bounds establish that uniform $AC^0$ circuits cannot compute Parity (Hahn, 2020). While soft-attention Transformers are strictly more expressive ($TC^0$), they remain bounded even with arbitrary precision if depth is fixed (Merrill & Sabharwal, 2023; Chiang, 2024). Part 2 (Rational Feature Solution): We construct a WFA $\mathcal{A}$ with $d = 2$ states that tracks the parity of the number of 1s. Let the state vector be $h_t \in \mathbb{R}^2$, where $h_t = (1,0)^\top$ represents an "Even" state and $h_t = (0,1)^\top$ represents an "Odd" state, and define (see Figure 3): *Initial State:* $\alpha = (1,0)^\top$; *Transitions:* For input token '0', $M_0 = I$. For input token '1', we set $M_1 = \begin{bmatrix} 0 & 1 \\ 1 & 0 \end{bmatrix}$.

The state update $h_t = M_{x_t} h_{t-1}$ performs exact modular arithmetic. If the number of ones is even, $h_T = (1,0)^\top$; if odd, $h_T = (0,1)^\top$. A linear readout $w = (0,1)^\top$ correctly identifies the parity. *Remark on Topology:* The matrix $M_1$ has determinant $-1$ (a reflection). While the standard Cayley parameterization in $d = 2$ is restricted to $SO(2)$ (rotations, det $+1$), this reflection can be realized in the Rational Transductor by augmenting the state to $d = 3$ (embedding the reflection as a rotation) or using a negative gain $g < 0$. $\square$

**Theorem 3** (Exact Modular Counting). *For any fixed integer $k \geq 2$, there exists a Rational Transductor with state dimension $d = k$ that exactly recognizes the language $L_k = \{x \in \{0,1\}^* : \#_1(x) \equiv 0 \pmod{k}\}$ for sequences of arbitrary length, whereas finite-depth Transformers cannot uniformly recognize $L_k$.*

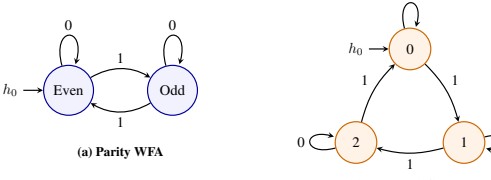

*Figure 3.* State tracking mechanisms for exact regular languages. (a) The Parity WFA uses a 2-state flip mechanism. (b) The Modulo-3 WFA generalizes this to a cyclic group structure.

*Proof sketch.* The negative result for standard Transformers follows from the fact that $MOD_k$ gates are not realizable in $AC^0$ for any $k \geq 2$ (Smolensky, 1987). For the Rational Transductor construction, we generalize the parity mechanism to the cyclic group $\mathbb{Z}_k$. We define a WFA with dimension $d = k$ where the basis vector $e_i$ represents the current count being $i \pmod{k}$. Let $M_0 = I_k$ (identity). Let $M_1$ be the cyclic permutation matrix where $(M_1)_{ij} = 1$ if $i \equiv (j+1) \pmod{k}$ and 0 otherwise. Since $M_1$ is a permutation matrix, it is orthogonal ($M^\top M = I$). Thus, using the Conservation Regime ($g = 1$) of the Rational Transductor, the state evolves unitarily, preserving the one-hot encoding exactly for any $T$. $\square$

## 4.3. Algebraic Completeness via Krohn-Rhodes

We can situate Rational Transductors within the broader hierarchy of sequence modeling architectures using the Krohn-Rhodes theorem.

**Theorem 4** (Representational Completeness). *Let $\mathcal{A}$ be any deterministic finite automaton. There exists a parameter setting for a Stacked Rational Transductor such that the model exactly simulates the state transitions of $\mathcal{A}$.*

*Proof sketch.* The Krohn-Rhodes theorem (Krohn & Rhodes, 1965) states that any finite automaton can be decomposed into a cascade (wreath product) of finite simple groups and aperiodic monoids. The Rational Transductor naturally implements this decomposition: *Group Components:* The Rational Feature Head, particularly with the Orthogonal/Cayley parameterization, implements the finite simple groups (permutations and cycles). *Aperiodic Components:* The Transformer backbone (via FFNs and attention heads) efficiently implements the aperiodic components (thresholds, resets, and feedback logic). $\square$

Thus, the architecture is not merely a heuristic ensemble, but a structurally complete neural realization of the algebraic components required to recognize any regular language.

**Extended Theoretical Characterizations.** We provide further structural results in Appendix C. We establish the Expressive Hierarchy (Proposition 19), proving that Rational Transductors strictly subsume both Rational Series and

Transformers while remaining strictly less expressive than general RNNs. We also prove Logical Completeness (Theorem 22), showing that under hard attention, the architecture corresponds exactly to Weighted Monadic Second-Order Logic (W-MSO[<]). Finally, we derive a Decomposition Characterization (Theorem 24) and a Circuit Complexity upper bound of $\mathsf{PNC}^1$ (Theorem 27), strictly separating the model from the $\mathsf{AC}^0$ limitations of standard Transformers and the sequential bottlenecks of P-complete RNNs. Table 1 summarizes the resulting expressivity landscape.

**From Representation to Induction.** Theorems 3 and 4 are statements about *representational capacity*—they guarantee the *existence* of parameters that exactly simulate any DFA. The critical bridge to *learnability* is provided by the optimization guarantees of Section 5. Specifically, Theorem 6 (Gradient Norm Preservation) establishes that the Cayley parameterization prevents vanishing/exploding gradients, while Theorem 7 (Time-Invariant Error Bounding) shows that the contractive regime ensures approximation error does not compound with sequence length. Together, these results confirm that the representational separation is not merely existential but *reachable by gradient descent*.

# 5. Theoretical Analysis of Learning

In this section, we provide a series of approximation, optimization, and learning guarantees for Rational Transductors. (For complete proofs and technical details for this section as well as additional learning theory results, we refer the reader to Appendix D.)

## 5.1. Universality of Random Features

We first prove that random rational features form a universal basis for sequential dependencies: a sufficiently wide random WFA can linearly reconstruct the state of any target WFA with probability one (Theorem 5).

**Theorem 5** (Universality of Random Features). *Let $\mathcal{A}_{rand}$ be a random WFA with sufficiently large state dimension $d$. With probability 1, the random feature states $\mathbf{h}_t$ form a universal basis that can linearly reconstruct the state of* any *target WFA up to length L, provided $d \geq |\Sigma|^L$.*

*Proof sketch.* The random WFA maps distinct input histories to random vectors in $\mathbb{R}^d$. If $d$ is large enough, these vectors are linearly independent with probability 1 (by generic property of random polynomials). Thus, there exists a linear projection $\mathsf{W}$ that maps these random states to the canonical states of the target automaton. $\square$

In Appendix D.1, we further establish a Uniform Spectral Approximation Bound (Theorem 32), showing that the approximation error scales as $O(C/\sqrt{d})$. Remarkably, this bound is governed solely by the random projection width $d$ and the target function's RKHS complexity $C$ (i.e., the squared norm in the Reproducing Kernel Hilbert Space induced by the kernel associated with the rational features), independent of sequence length. We also demonstrate a *Compactness Gap* (Proposition 33) to highlight the efficiency of training: while random features require a dimension $d \sim O(1/\epsilon^2)$ to solve tasks like Parity (Rahimi & Recht, 2007), learning the transitions yields an exact solution with just $d = 2$. This confirms that while our random regime analysis justifies the use of *near-identity initialization*, the *Differentiable* or *Learned Rational Feature* regime is strictly necessary for learning compact, efficient representations.

## 5.2. Optimization Stability

In this section, we present several favorable optimization guarantees for Rational Transductors. We first establish that the Rational Transductor possesses a structural immunity to the gradient explosion problem, distinguishing it from standard RNNs.

**Theorem 6** (Gradient Norm Preservation). *Consider the gradient of the loss $\mathcal{L}$ with respect to the hidden state $\mathbf{h}_t$, denoted $\boldsymbol{\delta}_t$. If the transitions satisfy the spectral constraint $\|\mathsf{M}_\sigma\|_2 \leq \gamma \leq 1$, then the propagated gradient norm satisfies:*

$$\|\boldsymbol{\delta}_{t-k}\|_2 \leq \gamma^k \|\boldsymbol{\delta}_t\|_2 + C$$

*Proof sketch.* The adjoint recurrence is given by $\boldsymbol{\delta}_{t-1} = \mathsf{M}_{x_t}^\top \boldsymbol{\delta}_t + \mathbf{v}_{t-1}$. Taking norms and applying the triangle inequality yields $\|\boldsymbol{\delta}_{t-1}\| \leq \|\mathsf{M}\|\|\boldsymbol{\delta}_t\| + \|\mathbf{v}\|$. By induction, the homogeneous term $\|\mathsf{M}\|^k$ either decays (if $\gamma < 1$) or remains bounded (if $\gamma = 1$). $\square$

Crucially, in the Conservation Regime ($\gamma = 1$), the gradient norm is strictly preserved (modulo injected terms), effectively solving the vanishing gradient problem for long-term dependencies.

This structural stability has a direct impact on learnability. The convergence rate of recurrent architectures is typically governed by the spectral gap $1 - \rho(\mathsf{M})$. Standard RNN training suffers from severe ill-conditioning as $\rho(\mathsf{M}) \to 1$ (the "vanishing/exploding gradient" boundary) (Pascanu et al., 2013). However, the result of Theorem 6 guarantees that for the Scaled Cayley parameterization, the optimization manifold remains well-conditioned even exactly at $\rho(\mathsf{M}) = 1$. This effectively pre-conditions the learning problem, allowing gradient descent to traverse the boundary of stability and learn infinite-horizon dependencies without divergence.

Finally, Theorem 36 (Appendix D.2) demonstrates that Deep Rational Injection prevents vanishing gradients. We further prove in Theorem 37 that the Hessian with respect to the transition matrix is bounded, ensuring the well-

*Table 1.* Theoretical comparison of capabilities between finite-depth Transformers and Rational Transductors. Transductors strictly expand expressivity to include all regular languages while sharing the fundamental limitation on context-free grammars.

| Task / Property | Transformer | Transductor | Theoretical Reason |
|---|---|---|---|
| Parity ($L_{\text{parity}}$) | ✗[†] | ✓ | Limited to $\mathsf{AC}^0/\mathsf{TC}^0$ |
| Modular Counting | ✗ | ✓ | Lack of cyclic state vs. WFA exactness |
| All Regular Languages | ✗ | ✓ | Star-free limitation vs. Rational completeness |
| Length Generalization | ✗ | ✓ | Positional drift vs. Time-invariant recurrence |
| All Context-Free Languages | ✗ | ✗ | Lack of unbounded memory stack |

[†] While soft-attention Transformers ($\mathsf{TC}^0$) can theoretically approximate Parity for fixed lengths via averaging, they cannot represent the solution *uniformly* for unbounded lengths without precision scaling or Chain-of-Thought, and empirically fail to generalize.

conditioned optimization landscape required for standard optimizers like AdamW.

### 5.3. Generalization and Robustness

In this section, we present fundamental learning guarantees for Rational Transductors, addressing length generalization, sample complexity, and stability. We first demonstrate that by relying on a recurrent state update, Rational Transductors learn a time-invariant transition rule. Crucially, the error of this rule does not compound over time but remains bounded, providing a theoretical explanation for the perfect length generalization observed in our experiments.

**Theorem 7** (Time-Invariant Error Bounding). *Let* $\mathsf{M}^*$ *be the true transition logic and* $\widehat{\mathsf{M}}$ *be the learned transition matrix. Assume the learned dynamics are contractive with* $\|\widehat{\mathsf{M}}\|_2 \le \gamma < 1$. *If the learned matrix approximates the true logic with error* $\|\widehat{\mathsf{M}} - \mathsf{M}^*\| \le \epsilon$, *then the deviation between the true state* $\mathbf{h}_t^*$ *and the Rational Transductor state* $\widetilde{\mathbf{h}}_t$ *is uniformly bounded for all* $t > 0$:

$$\sup_{t \ge 1} \|\mathbf{h}_t^* - \widetilde{\mathbf{h}}_t\| \le \frac{\epsilon C}{1 - \gamma}.$$

*Proof sketch.* Let $\mathbf{e}_t$ be the state error. The error evolution follows the recurrence $\mathbf{e}_t \approx \widehat{\mathsf{M}}\mathbf{e}_{t-1} + \Delta\mathbf{h}_{t-1}^*$. This defines a contractive map with a stable fixed point. Consequently, the error accumulates as a geometric series $\sum \gamma^k$, which converges to $(1 - \gamma)^{-1}$ regardless of sequence length. □

This result provides the theoretical justification for disabling explicit Positional Encodings (PEs) in algorithmic tasks, as the recurrent state itself provides a robust, length-independent anchor.

Furthermore, while standard generalization bounds for RNNs typically scale linearly or with the square root of the sequence length $T$, we show that for contractive Rational Transductors, the generalization gap is independent of $T$ (Theorem 39, Appendix D.3). We further characterize the sample complexity via the Hankel nuclear norm:

**Theorem 8** (Hankel-Rademacher Complexity). *The Empirical Rademacher complexity—a data-dependent measure of* *the richness of a function class that controls uniform generalization bounds—of the class of rational functions with Hankel nuclear norm bounded by* $r$ *scales as* $\widetilde{O}(r/\sqrt{N})$.

*Proof sketch.* This result exploits the duality between the Hankel nuclear norm and the spectral norm of the data matrix (Fliess' Theorem), applying Theorem 6 from (Balle & Mohri, 2017) (see Appendix D.3). □

This result establishes a fundamental link between spectral regularization and automata learning. Since the rank of the Hankel matrix corresponds exactly to the minimal state dimension $d_{\min}$ of the underlying WFA (Fliess' Theorem), controlling the nuclear norm—which our "Diagonal + Low Rank" parameterization effectively achieves—is rigorously equivalent to regularizing the *effective state dimension* of the latent automaton. This confirms that Rational Transductors generalize by discovering low-rank algebraic structures, distinguishing them from standard Transformers which often overfit to high-rank, spurious correlations.

Finally, Theorem 41 (Appendix D.3) guarantees that Rational Transductors are robust to small perturbations in input embeddings, ensuring stability against quantization noise or minor distribution shifts.

## 6. Empirical Validation

We validate our findings on synthetic tasks designed to probe the limitations of attention. We investigate two key claims: (1) whether Rational Transductors can solve $\mathsf{NC}^1$-complete tasks that are impossible for standard Transformers (the "Regular Gap"), and (2) whether the learned solutions generalize to unseen lengths. For complete experimental details, including hyperparameters, optimization settings, and statistical significance and reproducibility, refer to Appendix E.

### 6.1. The Regular Gap: Modulo Counting

**Task Setup.** We evaluated models on Modulo-5 Counting. We compared the Rational Transductor against a standard Transformer ($d_{model} = 32$, 2 layers). The Rational Transductor used a single rational head ($d_{\text{rat}} = 8$) initialized in

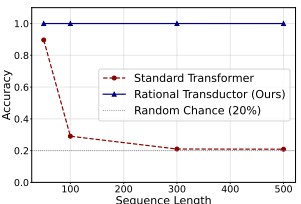 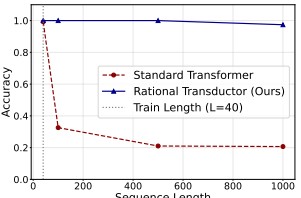

*(a)* **Regular Gap (Modulo-5).** The Standard Transformer (red) collapses to random chance (20%). The Rational Transductor (blue) maintains 100% accuracy.

*(b)* **Length Generalization.** Trained on $L$=40 (dashed), tested to $L$=1000. The Rational Transductor generalizes perfectly to 25× the training length.

*Figure 4.* Regular Gap and Length Generalization.

the *Strictly Orthogonal Regime* (Scaled Cayley with $g = 1$). Crucially, we disabled standard positional encodings, forcing the model to rely solely on the rational state for sequence tracking.

**Remark on State Space Models (SSMs).** We benchmark against Transformers and LSTMs to isolate the impact of linear recurrence versus attention or non-linear gating. While SSMs like Mamba (Gu & Dao, 2023) share our time-varying linear recurrence and theoretical PNC[1] capacity (under unitary initialization), the Rational Transductor serves as a *minimal theoretical proxy*. This allows us to analyze specific automata-theoretic mechanisms (e.g., orthogonal vs. stochastic transitions) without the confounding variables of Mamba's complex gating and block design.

**Results.** Figure 4a illustrates the performance. The Standard Transformer fails to learn a robust counting mechanism, achieving only partial success on training lengths ($L = 50$) and collapsing to random chance (20%) on longer sequences. In contrast, the Rational Transductor converges to 100% accuracy almost immediately. This confirms that the Rational Head successfully learns the underlying group-theoretic operation (cyclic permutation), validating the expressivity claims of Theorem 3.

## 6.2. Length Generalization and Time-Invariance

**Task Setup.** Models are trained solely on short sequences ($L_{\text{train}} = 40$) but evaluated on sequences up to $L_{\text{test}} = 1000$ (25× the training horizon). We use the Modulo-5 Counting task. The Standard Transformer uses standard Learned Absolute Positional Encodings to establish a baseline for the fundamental limits of the canonical architecture. The Rational Transductor disables positional encodings entirely.

Note that while relative encodings like RoPE or ALiBi allow length extrapolation, our analysis shows they are specific, fixed instances of Rational Features (input-independent unitary WFAs). Thus, Transformer+RoPE is theoretically subsumed by our framework. We therefore benchmark against the canonical absolute baseline to strictly isolate the con-

tribution of *learned, input-dependent* recurrence (semantic state tracking) versus standard attention (position tracking), unconfounded by manual inductive biases.

**Results.** As shown in Figure 4b, the Standard Transformer suffers from catastrophic positional drift. Once the sequence length exceeds the training horizon, the learned positional encodings are no longer valid, and performance drops to random guessing. The Rational Transductor, however, maintains perfect accuracy (> 99%) up to $L = 1000$.[1] This confirms that the model has learned an algebraically exact solution rather than an approximate one, consistent with the unitary parameterization ($\gamma = 1$).

Note that for the Standard Transformer baseline, the absolute positional embeddings are learned only on positions $1, \ldots, 40$ during training; for test lengths $L > 40$, the embeddings for positions $41, \ldots, L$ remain at their random initialization. This establishes the fundamental limit of the canonical architecture: without a length-invariant inductive bias, the model has no mechanism to extrapolate positional information beyond its training horizon.

## 6.3. Computational Efficiency

A core advantage of the Rational Transductor is its parallelizability. We benchmarked the inference latency (forward pass) of the sequence mixing layers in isolation (Rational Head vs. Self-Attention). This strictly isolates the algorithmic complexity of the recurrence ($O(\log T)$) versus the attention mechanism ($O(T^2)$), independent of the shared feed-forward blocks. We benchmarked on sequences ranging from $T = 128$ to $T = 32,768$ (Batch size $B = 1$). We compare against a Sequential RNN (linear scaling) and a FlashAttention Transformer (quadratic scaling) (Dao et al., 2022). All measurements were conducted on a single NVIDIA GPU (A100), averaged over 20–100 trials after a warm-up period.

**Results.** Figure 5 plots the latency. The Sequential RNN exhibits strict linear scaling ($O(T)$), becoming the bottleneck at extreme lengths (> 100ms for $T = 32$k). The Transformer scales efficiently for short lengths but hits a quadratic wall, exploding in latency and memory usage at extreme lengths. The Rational Transductor combines the best of both: for short sequences, it is competitive with the Transformer; for long sequences ($T > 512$), the parallel scan allows it to overtake the Sequential RNN, maintaining low latency even as $T$ reaches $32,768$. This empirically confirms the theoretical $O(\log T)$ parallel complexity.

---

[1] A minor precision artifact occurs near $L = 1000$: accumulated Float32 rounding errors in the orthogonal regime ($\gamma = 1$) can cause the state vectors to drift slightly from exact unit-norm after ~1000 matrix multiplications via the parallel scan. Using Float64 precision resolves this entirely, confirming it is an implementation artifact rather than a theoretical generalization failure.

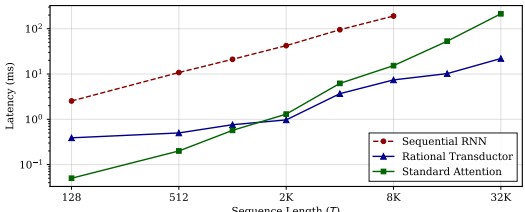

*Figure 5.* **Latency vs. Sequence Length.** The Rational Transductor (blue) leverages parallel associative scans to achieve sublinear scaling, outperforming the RNN on sequences longer than $T = 512$.

**Memory Complexity.** The parallel associative scan requires $O(T \cdot d^2)$ memory for storing the intermediate matrix products (where $d = d_{\text{rat}}$ is the rational state dimension), compared to $O(T^2)$ for standard self-attention. Since $d \ll T$ in practice (e.g., $d = 8$ in our experiments), the rational head contributes negligibly to peak memory relative to the attention layers.

### 6.4. Algorithmic Generalization: Long-Integer Addition

**Task Setup.** To test discrete, discontinuous logic, we evaluated models on Long-Integer Addition ($L$-digit numbers). This requires implementing a "Full Adder" state machine, where the carry bit must be generated (sum > 9), propagated (sum = 9), or killed (sum < 9). Models are trained on short numbers ($L \in [10, 40]$) and evaluated on lengths up to $L = 1000$. We used the *Universal Rational Transductor* configuration with a Stochastic component ($d_{rat} = 4$), parameterized via column-stochastic matrices (Softmax).

**Results.** Figure 6a shows the sequence-level accuracy. The Standard Transformer fails to generalize, dropping to 0% accuracy outside the training window ($L = 20$). Attention mechanisms struggle to maintain the hard sequential dependency of a carry bit over hundreds of steps. In contrast, the Universal Rational Transductor achieves 100% accuracy up to $L = 1000$. This confirms that by providing a diverse set of dynamic kernels (here, a Stochastic head), the model can autonomously learn complex algorithmic rules that require both infinite memory conservation and discrete state switching.

We further validate our framework with the quantitative accumulation experiment in Appendix E.1, where the Rational Transductor learns exact Base-2 integer evaluation to machine precision, while standard architectures fail due to saturation and attention noise (see Figure 6b).

## 7. Conclusion

We introduced *Rational Transductors*, a hybrid architecture that bridges the semantic flexibility of Transformers with the rigid state-tracking of formal languages. By augmenting self-attention with a linear, matrix-valued recur-

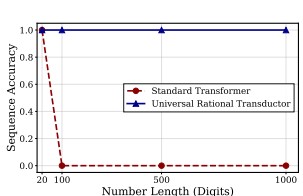

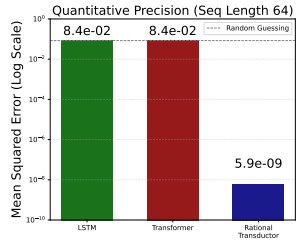

*(a)* **Long-Integer Addition.** The Standard Transformer (red) fails completely (0% at $L = 100$). The Universal Rational Transductor (blue) generalizes perfectly to $L = 1000$ digits.

*(b)* **Base-2 Evaluation.** Standard architectures (LSTM, Transformer) fail completely. The Rational Transductor learns the exact affine recurrence, achieving near-perfect precision.

*Figure 6.* Algorithmic generalization experiments.

rence—parameterized via our novel Scaled Cayley mechanism—we overcome the expressivity and learnability barriers inherent to standard Transformers. Our theoretical analysis aligns this framework with the Krohn-Rhodes decomposition, demonstrating that Transductors structurally capture the full hierarchy of regular languages, including the cyclic groups that $\text{AC}^0$ and $\text{TC}^0$ models fail to represent uniformly. Empirically, this yields solved length generalization: Rational Transductors achieve perfect accuracy on tasks like parity and modular addition where standard models fail, offering a rigorous path toward neuro-symbolic architectures that unify connectionist learning with algebraic reasoning. In particular, Rational Transductors generalize perfectly to $25\times$ the training horizon and solve 1000-digit addition with $100\%$ accuracy, while maintaining $O(\log T)$ parallel efficiency.

**Discussion and Scope.** This paper establishes a *foundational theory* for hybrid architectures that combine attention with linear recurrence. Our experimental validation focuses on synthetic tasks chosen to isolate the key theoretical properties (algebraic completeness, length generalization, state tracking). While we do not include large-scale NLP pretraining experiments, we emphasize that the architecture imposes *no constraints* on the Transformer backbone: the rational head is a modular addition that can be integrated into any existing Transformer. Scaling to full pretraining, in the spirit of Jamba (Lieber et al., 2024) or Mamba-3 (Lahoti et al., 2025), is the immediate next step. Beyond sequence modeling, we anticipate that the Rational Head's ability to learn exact group-theoretic operations will prove useful in program synthesis, formal verification, and any domain where rigid rule following must coexist with learned semantic representations. The combination of algebraic interpretability and gradient-friendly parameterization positions Rational Transductors as a principled building block for next-generation neuro-symbolic systems. A discussion of limitations and the capacity-complexity trade-off is provided in Appendix F.

## Acknowledgments

I thank Corinna Cortes and Will Merrill for very helpful comments on earlier drafts of this paper.

## Impact Statement

This work advances the theoretical understanding of sequence modeling efficiency. The proposed architecture improves parallel training capabilities ($O(\log T)$), which may reduce the energy footprint of training large-scale models. We do not foresee immediate negative societal consequences specific to this theoretical contribution.

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

# Contents of Appendix

# A. Theoretical Background: Weighted Automata and Rational Power Series

In this appendix, we place the architecture of Rational Transductors within the broader theoretical framework of Weighted Finite Automata (WFAs) (Mohri, 2009) and Rational Power Series (Salomaa & Soittola, 1978; Berstel & Reutenauer, 1988; Kuich & Salomaa, 1986). We define the specific class of series computed by our model and outline the fundamental theorems that guarantee their expressivity and learnability.

## A.1. Rational Power Series over a Field

Let $\Sigma$ be a finite alphabet and $\Sigma^*$ be the free monoid generated by $\Sigma$. A formal power series $S$ with coefficients in the field of real numbers $\mathbb{R}$ is a mapping $S : \Sigma^* \to \mathbb{R}$. The value of $S$ on a sequence $x \in \Sigma^*$ is denoted by $(S, x)$. The set of all such formal power series is denoted by $\mathbb{R}\langle\!\langle \Sigma^* \rangle\!\rangle$. The subset of *Rational Power Series*, denoted $\mathbb{R}^{\mathrm{rat}}\langle\!\langle \Sigma^* \rangle\!\rangle$, is the smallest subalgebra of $\mathbb{R}\langle\!\langle \Sigma^* \rangle\!\rangle$ containing all polynomials (series with finite support) that is closed under the following rational operations:

- **Sum:** $(S + T, x) = (S, x) + (T, x)$;

- **Cauchy Product:** $(S \cdot T, x) = \sum_{uv=x} (S, u)(T, v)$

- **Kleene Star (Closure):** $S^* = \sum_{n=0}^{\infty} S^n$, provided $(S, \epsilon) = 0$ to ensure convergence.

## A.2. Linear Representations and WFAs

A fundamental result in the theory of weighted automata is the Schützenberger representation theorem (Schützenberger, 1961), which establishes that rational series are exactly those recognizable by finite weighted automata.

**Definition 9** (Linear Representation). *A linear representation of dimension $d$ over $\mathbb{R}$ is a triple $(\boldsymbol{\alpha}, \{\mathsf{M}_\sigma\}_{\sigma \in \Sigma}, \boldsymbol{\beta})$, where:*

- $\boldsymbol{\alpha} \in \mathbb{R}^d$ *is the initial weight vector (column).*

- $\mathsf{M}_\sigma \in \mathbb{R}^{d \times d}$ *are the transition matrices for each $\sigma \in \Sigma$.*

- $\boldsymbol{\beta} \in \mathbb{R}^d$ *is the final weight vector (column).*

This representation computes a series $S$ defined by:

$$(S, x) = \boldsymbol{\beta}^\top \mathsf{M}_{x_T} \ldots \mathsf{M}_{x_1} \boldsymbol{\alpha}. \tag{5}$$

## A.3. Fundamental Results

**The Hankel Matrix and Fliess' Theorem.** A central tool in the analysis of rational series is the *Hankel matrix* $H_S$, an infinite matrix indexed by pairs of strings $(u, v) \in \Sigma^* \times \Sigma^*$, where the entry at $(u, v)$ is $(S, uv)$.

**Theorem 10** ((Fliess, 1974)). *A series $S$ is rational if and only if its Hankel matrix $H_S$ has finite rank. Furthermore, the rank of $H_S$ is equal to the dimension $d_{\min}$ of the minimal linear representation of $S$.*

**Minimization and Learning.** For any rational series, the minimal linear representation is unique up to a similarity transformation (see (Balle & Mohri, 2015) for a short proof and illustration). Furthermore, spectral learning frameworks (Balle & Mohri, 2012) demonstrate that rational series can be learned efficiently under specific rank conditions via the singular value decomposition of the Hankel matrix.

## A.4. Remarks on Terminology and Graph Interpretation

To clarify the relationship between our linear algebraic definition and standard automata theory, we provide the following remarks.

**Equivalence of Dimension and State Count.** The definition of a WFA via a linear representation of dimension $d$ is mathematically isomorphic to a Weighted Finite Automaton with exactly $d$ states.

- The dimension $d$ corresponds to the set of discrete states $Q = \{q_1, \ldots, q_d\}$.

- The entry $(\mathsf{M}_\sigma)_{ij}$ corresponds to the weight of the edge transitioning from state $q_j$ to state $q_i$ upon reading symbol $\sigma$.

- Consequently, Fliess' theorem can be equivalently stated as: the rank of the Hankel matrix equals the number of states in the minimal WFA recognizing the series.

**The State Vector as a Weight Distribution.** In the context of deep learning, the vector $h_t \in \mathbb{R}^d$ is often referred to as the "hidden state." In the automata theoretic view, this vector represents the *distribution of accumulated weights* over the $d$ states of the automaton at time $t$. Specifically, the $i$-th component $h_{t,i}$ is the sum of weights of all paths in the automaton ending at state $q_i$ given the input prefix $x_{1:t}$.

**Deterministic Computation in Vector Space.** While Weighted Finite Automata are not generally determinizable in the graph sense (i.e., transforming into an equivalent WFA with only one non-zero path per string) (Mohri, 2009), the linear recurrence $\mathbf{h}_t = \mathsf{M}_{x_t}\mathbf{h}_{t-1}$ constitutes a deterministic update in the vector space $\mathbb{R}^d$. This ensures that the Rational Transductor architecture remains deterministic and efficient to compute, despite the underlying WFA potentially representing non-deterministic weighted paths.

Finally, WFAs have been successfully used in a variety of applications, including speech recognition (Mohri, Pereira, and Riley, 2002). The OpenFST software library (Allauzen, Riley, Schalkwyk, Skut, and Mohri, 2007) provides a very general and efficient implementation of the representation and algorithms related to WFAs.

For an elegant introduction to finite automata and their theoretical connections, we refer the reader to (Perrin, 1990), and for a deeper mathematical analysis to (Eilenberg, 1974; 1976).

# B. Extended Architectural Analysis

## B.1. Positional Encodings as Rational Features

**Lemma 11** (Positional Encodings are Rational). *Standard Rotary Positional Embeddings (RoPE) are a special case of Rational Features where the transition matrices are input-independent unitary rotations.*

*Proof.* Standard sinusoidal and rotary embeddings are defined by frequencies $\Theta = \{\theta_i\}_{i=1}^{d/2}$. The encoding at position $t$ is typically constructed by rotating pairs of dimensions. Consider the block-diagonal rotation matrix R:

$$\mathsf{R} = \text{diag}(\mathsf{R}_{\theta_1}, \dots, \mathsf{R}_{\theta_{d/2}}), \quad \text{where } \mathsf{R}_{\theta_i} = \begin{pmatrix} \cos\theta_i & -\sin\theta_i \\ \sin\theta_i & \cos\theta_i \end{pmatrix}. \tag{6}$$

We define a WFA $\mathcal{A}$ with initial state $\boldsymbol{\alpha} = (1, 0, \dots, 1, 0)^\top$ and transition matrix $\mathsf{M}_\sigma = \mathsf{R}$ for all $\sigma \in \Sigma$. The state evolves as $\mathbf{h}_t = \mathsf{R}^t \boldsymbol{\alpha}$, which exactly matches the definition of sinusoidal positional encodings. Thus, RoPE is a specific instance of a Rational Feature head with input-independent, unitary transitions. $\square$

## B.2. Architectural Extensions

**Stacked Rational Transductors.**    While the canonical Rational Transductor architecture (Figure 1) uses a single rational head broadcast to all layers, the framework naturally admits a stacked generalization. A *Stacked Rational Transductor* of depth $K$ consists of a sequence of $K$ Rational Transductor blocks, where the output stream of the $k$-th block serves as the input to the $(k+1)$-th block. Formally, let $\mathbf{u}_t^{(k-1)}$ be the input vector to block $k$ at time $t$. The block computes a local rational state $\mathbf{h}_t^{(k)}$ and updates the residual stream:

$$\mathbf{h}_t^{(k)} = \mathsf{M}_{x_t}^{(k)} \mathbf{h}_{t-1}^{(k)} + \mathsf{V}^{(k)} \mathbf{u}_t^{(k-1)} \tag{7}$$

$$\mathbf{u}_t^{(k)} = \text{TransformerBlock}_k\left(\mathbf{u}_t^{(k-1)} + \mathsf{W}_{\text{proj}}^{(k)} \mathbf{h}_t^{(k)}\right) \tag{8}$$

where $\mathsf{M}^{(k)}$ is the transition logic specific to layer $k$.

**The Linear Collapse Property.**    A crucial theoretical observation guides our preference for the single-head design over the naive stack. In the regime where the inter-layer dependence is mediated by a linear map applied to the previous block's output (and no non-linearity is applied to the recurrent state), the cascade of WFAs is reducible.

**Proposition 12** (Reducibility of Cascaded WFAs). *A cascade of $K$ linear Weighted Finite Automata, where the state of automaton $k$ depends linearly on the state of automaton $k - 1$, is algebraically equivalent to a single WFA with a larger state space dimension $d_{total} = \sum_{k=1}^{K} d_k$.*

*Proof.* Consider two stacked states $\mathbf{h}^{(1)}$ and $\mathbf{h}^{(2)}$. The joint system update can be written as a block triangular matrix:

$$\begin{pmatrix} \mathbf{h}_t^{(1)} \\ \mathbf{h}_t^{(2)} \end{pmatrix} = \begin{pmatrix} \mathsf{M}^{(1)} & 0 \\ \mathsf{W}_{\text{inter}} & \mathsf{M}^{(2)} \end{pmatrix} \begin{pmatrix} \mathbf{h}_{t-1}^{(1)} \\ \mathbf{h}_{t-1}^{(2)} \end{pmatrix} \tag{9}$$

This block matrix defines a valid transition matrix $\mathsf{M}_{\text{joint}}$ for a single WFA. Thus, stacking linear recurrences does not strictly expand the class of representable functions beyond $\mathcal{T}_{\text{Rat}}$; it merely structures the transition matrix. $\square$

Despite this algebraic reducibility, there is a representation-theoretic benefit to the stacked parameterization. By enforcing the block-triangular structure inherent in the cascade, the model uses significantly fewer parameters to represent the same total state dimension, acting as a strong inductive bias for decomposable processes.

**Theorem 13** (Cascaded Parameter Efficiency). *Let $\mathcal{C}$ be a cascade of $K$ linear WFAs with state dimensions $d_1, \dots, d_K$. The number of parameters required to specify the transitions of the cascade is $O(|\Sigma| \sum_{k=1}^{K} d_k^2)$. In contrast, a generic (unconstrained) single WFA with the equivalent state dimension $D = \sum_{k=1}^{K} d_k$ requires $O(|\Sigma|(\sum_{k=1}^{K} d_k)^2)$ parameters. Thus, the stacked architecture enforces a sparsity constraint that reduces the sample complexity of learning by a factor of $O(K)$ when $d_k \approx d$.*

**(a) Rational Transductor (Wide)**   **(b) Non-Linear Stack (Deep)**

*Figure 7.* Architectural Comparison. (a) Wide Recurrence: The Rational Transductor computes a single high-dimensional state $h_t$ directly from the input via a parallel scan, injecting it into all layers. (b) Deep Recurrence: Stacked architectures (e.g., H3, Mamba) interleave recurrence, where Layer $k$ depends on the output of Layer $k-1$, reintroducing a sequential bottleneck during training.

**Non-Linear Stacking and Deep Recurrence.**   A natural question arises regarding the interaction between the rational features and the non-linear components of the Transformer. While our proposed architecture (Figure 1) uses a "sidecar" design where the rational states are strictly input-driven, one could alternatively construct a *Non-Linear Stacked Transductor*. In this variant, the input to the $k$-th rational head is not the original token embedding, but the *non-linear output* of the $(k-1)$-th Transformer block.

Unlike the linear case, non-linear stacking strictly increases the expressive capacity of the architecture per unit of state dimension. Because the transition dynamics of layer $k$ depend on the non-linear transformation of layer $k-1$, the system can realize functions that are not realizable by any single linear WFA of the aggregate dimension.

**Theorem 14** (Non-Linear Irreducibility). *There exist cascades of rational transductors with inter-layer non-linearities such that any single linear WFA capable of realizing the same transduction requires a state dimension $D \gg \sum_{k=1}^{K} d_k$.*

*Proof.* Consider a cascade of two WFAs ($K = 2$) with state dimensions $d_1, d_2 \geq 3$. Let the first WFA compute a state $h_t^{(1)} \in \mathbb{R}^{d_1}$ and the second compute $h_t^{(2)} \in \mathbb{R}^{d_2}$, where the input to the second WFA is modulated by the state of the first via a multiplicative interaction (e.g., attention or gating $u_t^{(2)} = u_t^{(1)} \otimes h_t^{(1)}$). To simulate this non-linear interaction with a single linear system, one must linearize the product state space, requiring a state vector isomorphic to the tensor product $h_t^{(1)} \otimes h_t^{(2)}$. The dimension of this linearized system is $D = d_1 \times d_2$. For all $d_1, d_2 \geq 2$ (and strictly for $d > 2$), the tensor product dimension exceeds the sum of dimensions ($d_1 d_2 \geq d_1 + d_2$, strict for $d > 2$). Thus, the non-linear cascade represents a function class that is more compact (linear vs quadratic in $d$) than any equivalent single linear WFA. $\square$

However, this expressivity comes at a steep computational cost. In our "Single Head" design, the states $\mathbf{h}_t$ are rational functions of the *input* $x$. Consequently, the entire state history for all layers can be pre-computed in parallel using a single pass of associative scans ($O(\log T)$ depth). In contrast, in the Non-Linear Stack, the input to the $k$-th recurrence $\mathbf{u}_t^{(k-1)}$ depends on the attention output of the previous layer. This introduces a *layer-wise sequential dependency*: the parallel scan for layer $k$ cannot commence until layer $k-1$ is fully computed. For a model with $L$ layers, the parallel complexity scales as $O(L \log T)$.[2] For deep foundation models ($L \approx 96$) (Brown et al., 2020), this serialization reintroduces a significant training bottleneck. Our architecture thus prioritizes parallel efficiency, opting to capture complex dependencies via a wider, input-driven rational state rather than a deep, serial one.

---

[2]While theoretical circuit constructions can potentially parallelize deep non-linear networks to $O(\text{polylog } T)$ depth (e.g., via divide-and-conquer on the composition of layers), standard layer-wise implementations in deep learning frameworks enforce a sequential dependency of depth $L$, making $O(L \log T)$ the practical latency lower bound.

# C. Expressivity and Complexity

In this section, we analyze the expressivity and generalization capabilities of Rational Transductors. We summarize our main findings in Table 1, which highlights that Transductors strictly generalize standard Transformers to capture all regular languages and $NC^1$ complexity.

**Theoretical Setup.** Unless otherwise stated, all expressivity results assume exact or bounded-precision arithmetic with error tolerance independent of sequence length. For lower bounds, we define "Standard Transformers" as *uniform families of fixed-depth architectures without auxiliary memory* (no Chain-of-Thought). Under these constraints, Transformers are limited to $AC^0$ (hard attention) (Hahn, 2020) or $TC^0$ (soft attention) (Merrill & Sabharwal, 2023; Chiang, 2024), preventing them from solving sequential problems like Parity which require depth scaling with input length. Note that while this section focuses on representational capacity, we explicitly address numerical stability and optimization dynamics in Section 5.

**Interpretation of Results.** Table 1 illustrates that Transductors occupy just the *right* zone of expressivity. By integrating the WFA head, they overcome the fundamental inability of standard Transformers (restricted to $AC^0$ under hard attention, or $TC^0$ under soft attention) to *uniformly* handle periodic and sequential logic (such as Parity and $MOD_k$), effectively solving the *Regular Gap*. Crucially, however, Transductors do not attempt to solve the *Context-Free Gap*; like standard Transformers, they lack the unbounded stack required for tasks like hierarchical bracket matching. This deliberate design choice preserves the $O(\log T)$ parallel training efficiency that general RNNs or stack-augmented models sacrifice.

## C.1. Positional Encodings and Generalization

We first show that our framework strictly generalizes modern positional encoding schemes.

**Lemma 15** (Positional Encodings are Rational). *Over the field of real numbers $\mathbb{R}$, let $P \in \mathbb{R}^{T \times d}$ be the matrix of Rotary Positional Embeddings (RoPE) or sinusoidal encodings. There exists a Weighted Finite Automaton $\mathcal{A} = (\Sigma, d, \boldsymbol{\alpha}, \{M_\sigma\})$ such that for any position $t$, the rational feature vector $\mathbf{h}_t$ is exactly the positional encoding at index $t$, independent of the input tokens.*

*Proof.* Standard sinusoidal and rotary embeddings are defined by frequencies $\Theta = \{\theta_i\}_{i=1}^{d/2}$. The encoding at position $t$ is typically constructed by rotating pairs of dimensions. Consider the block-diagonal rotation matrix $R$:

$$R = \begin{pmatrix} R_{\theta_1} & 0 & \dots & 0 \\ 0 & R_{\theta_2} & \dots & 0 \\ \vdots & \vdots & \ddots & \vdots \\ 0 & 0 & \dots & R_{\theta_{d/2}} \end{pmatrix}, \quad \text{where } R_{\theta_i} = \begin{pmatrix} \cos\theta_i & -\sin\theta_i \\ \sin\theta_i & \cos\theta_i \end{pmatrix} \tag{10}$$

We define a WFA $\mathcal{A}$ as follows:

- Let the initial state $\boldsymbol{\alpha} = (1, 0, 1, 0, \dots, 1, 0)^\top \in \mathbb{R}^d$.

- Let the transition matrix $M_\sigma = R$ for all $\sigma \in \Sigma$. (The transition is input-independent).

The rational feature vector at time $t$ evolves as $\mathbf{h}_t = M_{x_t} \dots M_{x_1} \boldsymbol{\alpha} = R^t \boldsymbol{\alpha}$. Since $R$ is block-diagonal, we can analyze the evolution of the $i$-th pair of dimensions independently:

$$\begin{pmatrix} h_{t,2i-1} \\ h_{t,2i} \end{pmatrix} = \begin{pmatrix} \cos\theta_i & -\sin\theta_i \\ \sin\theta_i & \cos\theta_i \end{pmatrix}^t \begin{pmatrix} 1 \\ 0 \end{pmatrix} = \begin{pmatrix} \cos(t\theta_i) \\ \sin(t\theta_i) \end{pmatrix}. \tag{11}$$

This matches exactly the definition of sinusoidal positional encodings. Thus, standard positional information is a special case of rational features where the transition matrices are unitary and input-independent. While RoPE is typically applied as a modulation within the attention mechanism, this lemma demonstrates that the underlying state tracking mechanism is representable by a WFA. $\square$

This lemma implies that Transductors start with the full capability of standard Transformers (via PE) but extend it by allowing the transition matrices $M_\sigma$ to be *input-dependent*, enabling the tracking of semantic states rather than just wall-clock time.

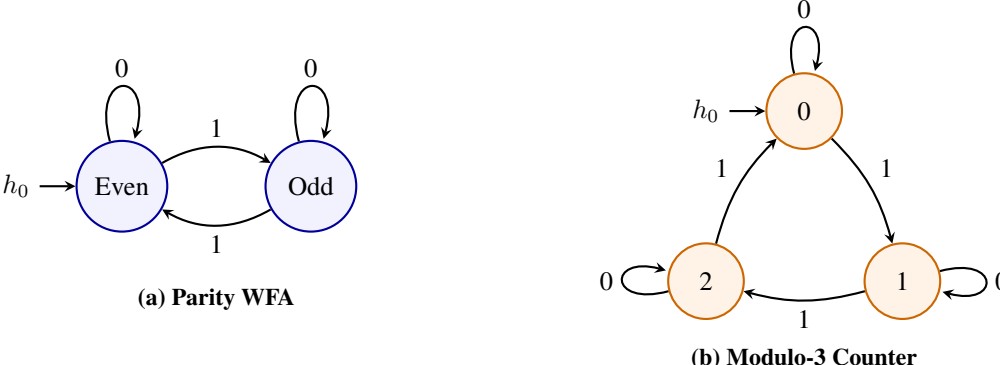

**(a) Parity WFA**

**(b) Modulo-3 Counter**

*Figure 8.* State tracking mechanisms for exact regular languages. **(a)** The Parity WFA uses a 2-state flip mechanism to track $L_{\text{parity}}$. **(b)** The Modulo-3 WFA generalizes this to a cyclic group structure to solve $L_k$ for $k = 3$. Input '0' acts as the Identity I (self-loop), while input '1' acts as a permutation.

The construction in Lemma 15 formalizes the "RoPE Trick" used in recent linear attention and SSM architectures (Lahoti et al., 2025; Yang et al., 2025b). However, while RoPE has traditionally been viewed as a positional modulation, this result provides the first rigorous proof that it constitutes a specific *Rational Inductive Bias*. This formalization explains precisely why RoPE fails to generalize length on algorithmic tasks: it constrains the recurrence to track *input-independent* relative position, rather than the *input-dependent* state required for semantic generalization.

### C.2. Expressive Separations: Parity and Counting

**Theorem 16** (The Parity Gap). *Let $L_{parity}$ be the language of binary strings $x \in \{0, 1\}^*$ containing an odd number of $1$s.*

1. *A standard Transformer with fixed depth, hard attention, and* bounded-precision arithmetic *cannot uniformly recognize $L_{parity}$ for input sequences of unbounded length.*

2. *There exists a Rational Transductor with state dimension $d = 2$ that recognizes $L_{parity}$ with $100\%$ accuracy for any length $T$.*

*Proof.* Part 1 (Transformer Limitation): Theoretical lower bounds establish that uniform $\text{AC}^0$ circuits (Hard Attention) cannot compute Parity (Hahn, 2020). While soft-attention Transformers are strictly more expressive, recent analysis proves they remain bounded within $\text{TC}^0$ (Threshold Circuits) even with arbitrary precision, provided they operate in fixed depth (Merrill & Sabharwal, 2023; Chiang, 2024). Turing completeness is only achievable when the model is permitted to generate intermediate *Chain-of-Thought* tokens, effectively using the context as a read-write tape (Pérez et al., 2021; Merrill & Sabharwal, 2024). Finite-depth models without such scratchpads remain bounded in computational power and, crucially, lack the inductive bias to learn robust algorithmic solutions (Merrill et al., 2022).

Part 2 (Rational Feature Solution): We construct a WFA $\mathcal{A}$ with $d = 2$ states that tracks the parity of the number of $1$s (see Figure 8(a)). Let the state vector be $\mathbf{h}_t \in \mathbb{R}^2$, where $\mathbf{h}_t = (1, 0)^\top$ represents an "Even" state and $\mathbf{h}_t = (0, 1)^\top$ represents an "Odd" state.

- Initial State: $\boldsymbol{\alpha} = (1, 0)^\top$ ($0$ is an even number).

- Transitions: For input token '0', the count does not change. We set $\mathsf{M}_0 = I = \begin{pmatrix} 1 & 0 \\ 0 & 1 \end{pmatrix}$. For input token '1', the parity flips. We set $\mathsf{M}_1 = \begin{pmatrix} 0 & 1 \\ 1 & 0 \end{pmatrix}$. (Note: This matrix has determinant $-1$. While standard Cayley parameterizations in $d = 2$ are restricted to $SO(2)$ (determinant $+1$), this reflection can be realized in the Rational Transductor by augmenting the state to $d = 3$ (embedding the reflection as a rotation).)

The state update $\mathbf{h}_t = \mathsf{M}_{x_t} \mathbf{h}_{t-1}$ performs exact modular arithmetic. If the number of ones is even, $\mathbf{h}_T = (1, 0)^\top$; if odd, $\mathbf{h}_T = (0, 1)^\top$. The Rational Transductor injects this $\mathbf{h}_T$ into the final layer. A simple linear classifier (readout head)

$\mathbf{w} = (0,1)^\top$ can then perfectly classify the sequence as $y = \text{sign}(\mathbf{w}^\top \mathbf{h}_T)$. This holds for any sequence length $T$, proving the claim. $\qquad\square$

Note that the construction in Part 2 explicitly relies on the learned parameterization (Regime II), where we can set $\mathsf{M}_\sigma$ to specific permutation matrices.

**Theorem 17** (Exact Modular Counting). *For any fixed integer $k \geq 2$, there exists a Rational Transductor with state dimension $d = k$ that exactly recognizes the language*

$$L_k = \{x \in \{0,1\}^* : \#_1(x) \equiv 0 \pmod{k}\}$$

*for sequences of arbitrary length, whereas finite-depth Transformers cannot uniformly recognize $L_k$.*

*Proof.* The negative result for standard Transformers follows from the fact that $MOD_k$ gates are not realizable in $\text{AC}^0$ for any $k \geq 2$ (Smolensky, 1987). For the Rational Transductor construction, we generalize the parity mechanism to the cyclic group $\mathbb{Z}_k$. We define a WFA with dimension $d = k$ where the basis vector $\mathbf{e}_i$ represents the current count being $i \pmod{k}$ (see Figure 8(b)).

- Initial State: $\boldsymbol{\alpha} = \mathbf{e}_0 = (1, 0, \ldots, 0)^\top$.

- Transitions: Let $\mathsf{M}_0 = I_k$ (identity). Let $\mathsf{M}_1$ be the cyclic permutation matrix where $(\mathsf{M}_1)_{ij} = 1$ if $i \equiv (j+1) \pmod{k}$ and 0 otherwise.

The state update $\mathbf{h}_t = \mathsf{M}_{x_t} \mathbf{h}_{t-1}$ ensures that if the current count is $j$, reading a '1' moves the probability mass to $(j+1) \pmod{k}$. Thus, $\mathbf{h}_T = \mathbf{e}_r$ where $r = \#_1(x) \pmod{k}$. A linear readout $\mathbf{w} = \mathbf{e}_0$ correctly identifies if $x \in L_k$. $\qquad\square$

*Note:* As with Parity, this capability assumes the learned regime, where the model can discover and maintain orthogonal transition matrices.

**Theorem 18** (Exact Arithmetic Evaluation). *Let $val_b: \Sigma^* \to \mathbb{N}$ be the function that interprets a string $x \in \{0, \ldots, b-1\}^*$ as a number in base $b$ (e.g., for binary $b = 2$, "101" $\mapsto$ 5). There exists a Rational Transductor with state dimension $d = 2$ that computes $val_b(x)$ exactly for sequences of arbitrary length (assuming computation over $\mathbb{R}$).*

*Proof.* This function corresponds to the Horner scheme for polynomial evaluation. We construct a WFA with state vector $\mathbf{h}_t = [v_t, 1]^\top$, where $v_t$ is the current accumulated value.

- **Initial State:** $\boldsymbol{\alpha} = [0, 1]^\top$.

- **Transitions:** For a digit $\sigma \in \{0, \ldots, b-1\}$, the update rule is $v_t = b \cdot v_{t-1} + \sigma$. This is an affine transformation representable by the matrix:

$$\mathsf{M}_\sigma = \begin{pmatrix} b & \sigma \\ 0 & 1 \end{pmatrix}. \tag{12}$$

The recurrence yields $\mathbf{h}_t = \mathsf{M}_{x_t} \mathbf{h}_{t-1}$. The first component of $\mathbf{h}_T$ exactly holds $\sum_{t=1}^{T} x_t \cdot b^{T-t}$. A linear readout $\mathbf{w} = [1, 0]^\top$ retrieves the integer value. Standard Transformers with bounded activations or attention scores cannot represent this unbounded growth function exactly, nor can standard RNNs with saturating non-linearities (e.g., $\tanh$). $\qquad\square$

**Generalization.** This result illustrates a capability distinct from the finite-state logic of Parity or Modulo-$k$. As shown in prior work on the expressivity of weighted automata (Cortes & Mohri, 2000), WFAs can represent broad families of functions mapping strings to numerical values, including polynomial evaluations and probabilistic distributions. The Rational Transductor inherits this capacity to model quantitative, unbounded dependencies that are inaccessible to architectures constrained by saturation or bounded precision.

**Recognition of Context-Free Languages.** While we emphasized the distinction between Regular and Context-Free capabilities in Table 1, the ability to perform exact arithmetic (Theorem 18) theoretically bridges this gap. As established in (Cortes & Mohri, 2000), the support of rational power series over infinite fields can characterize certain Context-Free Languages, including palindromes and Dyck paths, by leveraging arithmetic "counting" mechanisms to enforce structural constraints (e.g., mapping balanced nested structures to a zero-sum weight). Thus, Rational Transductors possess the latent capacity to "recognize" these languages through quantitative embedding, though robust learning of such unbounded arithmetic solutions in practice requires high precision and specific inductive biases.

### C.3. The Expressive Hierarchy

We can situate Rational Transductors within the broader hierarchy of sequence modeling architectures. The Rational Transductor framework occupies a precise theoretical niche, strictly generalizing both Rational Power Series and standard Transformers.

**Proposition 19** (Expressive Hierarchy). *Let $\mathcal{F}_{Rat}$, $\mathcal{F}_{TF}$, $\mathcal{F}_{RT}$, and $\mathcal{F}_{RNN}$ denote the classes of functions computable by Rational Power Series (linear WFAs), finite-depth Transformers, Rational Transductors, and general Recurrent Neural Networks, respectively. The following strict inclusions hold:*

$$(\mathcal{F}_{Rat} \cup \mathcal{F}_{TF}) \subsetneq \mathcal{F}_{RT} \subsetneq \mathcal{F}_{RNN}. \tag{13}$$

*Proof.* We establish each inclusion and its strictness separately.

1. Rational Series ($\mathcal{F}_{Rat} \subsetneq \mathcal{F}_{RT}$). Any rational series is computable by a WFA (Schützenberger, 1961). A Rational Transductor recovers this exactly by learning the corresponding transitions $\{M_\sigma\}$ and setting the Transformer layers to perform a fixed linear readout. The inclusion is strict because Transductors can apply non-linear operations (e.g., softmax attention, layer normalization) to the sequence of rational states, whereas $\mathcal{F}_{Rat}$ is limited to linear transductions.

2. Transformers ($\mathcal{F}_{TF} \subsetneq \mathcal{F}_{RT}$). A Rational Transductor recovers a standard Transformer if the rational projection weights are set to zero ($W_{proj} = 0$). The strictness is proven by Theorem 16 and Theorem 17: Transductors can uniformly recognize Parity and Modular Counting languages, whereas standard Transformers (limited to $AC^0$) cannot.

3. Recurrent Neural Networks ($\mathcal{F}_{RT} \subsetneq \mathcal{F}_{RNN}$). Rational Transductors are a specific instance of RNNs where the state transition is strictly linear ($h_t = M_{x_t} h_{t-1}$). General RNNs allow for non-linear state dynamics (e.g., $\tanh$), which encompass linear updates in the small-signal regime (linear region of the activation). The inclusion is strict because the linear state dynamics of the Transductor admit a parallel-prefix decomposition, limiting them to $PNC^1$. In contrast, general RNNs with non-linear activations and unbounded precision are P-complete (sequentially strictly harder). While linear Transductors can represent specific bounded Context-Free Languages (such as Boolean Formula Evaluation) via their $NC^1$ capacity (Huang et al., 2025), they lack the unbounded stack required to recognize *all* Context-Free Languages (e.g., Dyck-$k$ of arbitrary depth) uniformly. □

**Algebraic Completeness via Krohn-Rhodes.** From the perspective of algebraic automata theory, the Krohn-Rhodes theorem establishes that any finite state machine can be decomposed into a cascade of simple groups (counters) and aperiodic monoids (threshold logic). Standard self-attention is known to be limited to the aperiodic component (star-free languages) (Hahn, 2020). By incorporating linear recurrence, which naturally implements cyclic group operations (as seen in Theorem 17), Transductors effectively recover the group-theoretic component. Thus, Transductors provide a structurally complete architecture capable of modeling both the periodic and aperiodic sub-structures of all regular languages.

**Descriptive Complexity and MSO.** In the framework of descriptive complexity, standard Transformers are often associated with First-Order Logic (FO[<]), which describes star-free regular languages but fails to capture modulo counting quantifiers. In contrast, Weighted Finite Automata are intimately linked to Monadic Second-Order Logic (MSO[<]) over fields. By bridging these architectures, Transductors effectively lift the expressivity of the model from the limitations of first-order predicates to the fuller expressive range of monadic second-order logic on sequences.

**Proposition 20** (Effective Capacity Increase). *For any fixed horizon $L$, the pseudo-dimension of Rational Transductors strictly exceeds that of finite-depth Transformers with the same hidden dimension, due to the ability to linearly separate histories that are indistinguishable under attention-only architectures.*

*Proof.* Let $\mathcal{H}_{TF}$ and $\mathcal{H}_{RT}$ be the hypothesis classes of finite-depth Transformers and Transductors. Consider the set of

all binary sequences of length $L$, $S_L = \{0, 1\}^L$, and the subset of labelings defined by $y(x) = \text{Parity}(x)$. A Rational Transductor with $d = 2$ can realizably separate $S_L$ according to parity labelings (Theorem 16). Conversely, for fixed model size, there exists a length $L$ such that a standard Transformer ($\text{AC}^0$) cannot compute Parity (Furst et al., 1984), failing failing to realize the parity labeling on $S_L$. Since $\mathcal{H}_{\text{TF}} \subseteq \mathcal{H}_{\text{RT}}$ and Transductors can realize dichotomies (parity) that Transformers cannot, $\text{Pdim}(\mathcal{H}_{\text{RT}}) > \text{Pdim}(\mathcal{H}_{\text{TF}})$. $\qquad\square$

**Theorem 21** (Representational Completeness via Krohn-Rhodes). *Let $\mathcal{A}$ be any deterministic finite automaton with state set $Q$ and input alphabet $\Sigma$. There exists a parameter setting $\theta$ for a* Stacked *Rational Transductor (see Section B.2) of finite depth $L$ and width $d$ such that the model exactly simulates the state transitions of $\mathcal{A}$. Specifically, the architecture admits a parameterization that realizes the Krohn-Rhodes decomposition $(S_L \wr \cdots \wr S_1) \wr (R_L \wr \cdots \wr R_1)$, where the Rational Feature Heads implement the simple groups $S_i$ and the Transformer layers implement the cascading feedback functions required by the wreath product.*

*Proof.* The Krohn-Rhodes theorem states that any finite automaton can be decomposed into a cascade (wreath product) of finite simple groups and aperiodic monoids (resets) (Krohn & Rhodes, 1965). We construct a parameterization of the Rational Transductor that physically instantiates this cascade.

**1. The Wreath Product Structure.** A wreath product $M_2 \wr M_1$ is driven by a feedback function $\phi : Q_1 \times \Sigma \to \text{End}(Q_2)$, where the update rule for the second machine depends on the state of the first:

$$q_t^{(2)} = q_{t-1}^{(2)} \cdot \phi(q_{t-1}^{(1)}, x_t). \tag{14}$$

**2. Structural Mapping.** We map the cascade layers $1 \ldots L$ to the Transductor layers.

- **Group Components ($S_i$):** The Rational Feature Head at layer $i$ is parameterized to implement the group operations. For a simple group $G$, we set the transition matrices $\mathsf{M}_\sigma$ to be the permutation matrices representing the group elements.

- **Aperiodic/Feedback Components ($\phi$):** The Transformer block between layer $i$ and $i + 1$ implements the feedback function $\phi$. The input to layer $i + 1$'s head is $u_t = \text{FFN}(\text{LayerNorm}(z_t^{(i)}))$.

**3. Exact Implementation of Feedback Logic.** The function $\phi$ is a map over a finite domain (discrete states $Q_1$ and alphabet $\Sigma$). It is a standard result that a Feed-Forward Network (FFN) with ReLU activations and sufficient width can *exactly* represent any function over a finite boolean domain (e.g., implementing arbitrary logic gates or look-up tables). Thus, there exists a setting of the FFN weights such that:

$$\mathsf{M}_t^{(i+1)} = \text{Proj}(\text{FFN}(\text{Embed}(q_{t-1}^{(1)}) + \text{Embed}(x_t))) \tag{15}$$

exactly recovers the transition operator required by the wreath product logic $\phi(q_{t-1}^{(1)}, x_t)$.

**Conclusion.** Since the architecture contains components capable of representing both the algebraic primitives (via linear recurrence matrices) and the arbitrary Boolean glue logic (via FFNs), there exists a parameter configuration that exactly simulates the full Krohn-Rhodes cascade. $\qquad\square$

**Algebraic Division of Labor.** This structural alignment highlights a natural division of labor within the architecture. The Rational Feature Head is uniquely suited to implement the *permutation* components (finite simple groups) of the decomposition via orthogonal recurrence matrices—precisely the structures that standard self-attention fails to represent uniformly. Meanwhile, the Transformer backbone (via FFNs and attention heads) efficiently implements the *aperiodic* components (reset logic, thresholds) and the *cascading* dependencies required to glue the decomposition together.

**Significance: A Constructive Algebraic Completion.** This result offers a fundamental algebraic justification for the Rational Transductor architecture, moving beyond simple expressive capacity arguments. It is established that standard self-attention is limited to recognizing star-free languages, which algebraically correspond to aperiodic monoids (threshold logic and resets). By augmenting this aperiodic component with a linear recurrence capable of implementing finite simple groups (permutations and cycles), the Rational Transductor effectively physically instantiates the Krohn-Rhodes decomposition of finite automata. In this view, the "Deep Rational Injection" mechanism functions as the wreath product operator, cascading the cyclic state information (from the group component) into the aperiodic logic (of the attention component). Thus, the

architecture is not merely a heuristic ensemble, but a structurally complete neural realization of the fundamental algebraic components required to recognize *any* regular language, explicitly repairing the specific group-theoretic deficiency of the Transformer.

**Theorem 22** (Logical Characterization via Weighted MSO). *Let W-MSO$[<]$ be the Weighted Monadic Second-Order Logic over the commutative semiring $\mathbb{K}$ (in this work, the field $\mathbb{K} = (\mathbb{R}, +, \times)$). Under the restriction of* hard attention *(which implements* FO$[<]$ *selection), the class of functions representable by Rational Transductors $\mathcal{F}_{RT}$ strictly subsumes the class of functions definable in W-MSO$[<]$. That is:*

$$\mathcal{F}_{RT}[HardAttn] = [\![W\text{-}MSO[<]]\!]. \tag{16}$$

*Proof.* We establish the equality by mutual inclusion.

**1. Lower Bound ($[\![\mathbf{W\text{-}MSO}]\!] \subseteq \mathcal{F}_{\mathbf{RT}}$).** A fundamental result by Droste & Gastin (2007) establishes that for any formula $\phi \in$ W-MSO$[<]$ over a field, there exists a Weighted Finite Automaton (WFA) $\mathcal{A}_\phi$ that computes the same series. Since the Rational Feature Head of a Transductor can implement any WFA (by learning the transition matrices $\mathsf{M}_\sigma$), and the Transformer layers can implement the linear readout (by setting $\mathsf{W}_{\text{proj}}$ to inject the final state and attention to identity), every W-MSO formula is realizable by a Rational Transductor.

**2. Upper Bound ($\mathcal{F}_{\mathbf{RT}} \subseteq [\![\mathbf{W\text{-}MSO}]\!]$).** We show that the computation of a Rational Transductor is definable in W-MSO.

- **Recurrence:** The linear update $\mathbf{h}_t = \mathsf{M}_{x_t}\mathbf{h}_{t-1}$ is a regular recurrence, which is known to be definable in MSO (specifically, the relation "state at $t$ is $q$" is MSO-definable).

- **Hard Attention:** Hard attention performs a selection $y_t = x_k$ where $k = \text{argmax}_j(q_t \cdot k_j)$. The $\text{argmax}$ and indexing operations are First-Order (FO$[<]$) definable relations.

- **Composition:** Since FO$[<] \subset$ MSO$[<]$, the composition of the recurrence (MSO) and the attention mechanism (FO) remains within W-MSO.

Thus, the entire input-output mapping of the hard-attention Rational Transductor can be described by a single Weighted MSO formula. $\square$

**Soft Attention and Extended Expressivity.** We note that standard Transformers use *soft attention*, which strictly exceeds the expressivity of FO$[<]$ and can approximate specific Context-Free Languages like Dyck-1 ($a^n b^n$) (Merrill & Sabharwal, 2023; Chiang, 2024). Consequently, a Rational Transductor with soft attention theoretically exceeds the W-MSO upper bound. However, the hard-attention equivalence (Theorem 22) serves to formalize the specific contribution of the Rational Head: it provides the *monadic quantifiers* (state tracking) that are structurally absent in the attention mechanism, regardless of precision.

**Remark on Logical Completeness (Weighted MSO).** This algebraic perspective naturally extends to descriptive complexity. It is a classical result that Weighted Finite Automata are expressively equivalent to *Weighted Monadic Second-Order Logic* (W-MSO$[<]$) (Droste & Gastin, 2007). Standard Transformers are known to correspond to First-Order Logic (FO$[<]$), which effectively captures the "aperiodic" component of language but fails to express the "group" component (modulo counting). By structurally integrating a WFA (which captures the full power of MSO on sequences) with Attention (FO), the Rational Transductor architecture effectively bridges the gap between First-Order and Second-Order logic. This implies the model is *logically complete* for quantitative regular properties, contrasting with standard Transformers which are strictly limited to first-order expressivity.

## C.4. Structural Characterization

We can rigorously characterize the class of functions computable by Rational Transductors $\mathcal{F}_{\mathsf{RT}}$ as the exact composition of the Transformer class with the class of Vector-Valued Rational Functions.

**Definition 23** (Vector-Valued Rational Functions). *Let $\Sigma$ be an input alphabet. A function $\Phi: \Sigma^* \to (\mathbb{R}^d)^*$ is a Vector-Valued Rational Function if it is realizable by a Weighted Finite Automaton (WFA). That is, there exists a linear representation $(\boldsymbol{\alpha}, \{\mathsf{M}_\sigma\}_{\sigma \in \Sigma})$ such that for any input $x = (x_1, \ldots, x_T)$, the output is the sequence of state vectors $\Phi(x) = (\mathbf{h}_1, \ldots, \mathbf{h}_T)$ defined by the recurrence $\mathbf{h}_t = \mathsf{M}_{x_t}\mathbf{h}_{t-1}$ (with $\mathbf{h}_0 = \boldsymbol{\alpha}$). We denote this class as $\mathcal{T}_{Rat}$.*

**Theorem 24** (Decomposition Characterization). *Let $\mathcal{T}_{Rat}$ be the class of vector-valued rational functions defined above. Let $\mathcal{F}_{TF}$ be the class of functions computable by finite-depth Transformers. The class of Rational Transductor functions is exactly the composition of these two classes:*

$$\mathcal{F}_{RT} = \mathcal{F}_{TF} \circ \mathcal{T}_{Rat} \tag{17}$$

*That is, a function $F$ is a Rational Transductor if and only if $F(x) = G(\Phi(x))$ for some $G \in \mathcal{F}_{TF}$ and $\Phi \in \mathcal{T}_{Rat}$.*

*Proof.* Direction 1 ($\mathcal{F}_{RT} \subseteq \mathcal{F}_{TF} \circ \mathcal{T}_{Rat}$): We show that the Deep Rational Injection mechanism can be simulated by a standard Transformer given access to the rational states. A Rational Transductor computes layers via $\mathbf{z}_t^{(l+1)} = \text{TF}_l\left(\mathbf{z}_t^{(l)} + \mathsf{W}_{\text{proj}}^{(l)}\mathbf{h}_t\right)$. Consider a standard Transformer $G$ taking the concatenated input $u_t = [\mathbf{x}_t; \mathbf{h}_t]$. A standard Transformer can simulate the Transductor's deep injection by: (1) Dedicating a subspace of its residual stream to copy $\mathbf{h}_t$ forward to all layers (using identity attention/FFN weights); and (2) At each layer $l$, applying the linear operation corresponding to $\mathsf{W}_{\text{proj}}^{(l)}$ to this subspace and adding it to the processing stream. Thus, any function computed by a Transductor is computable by a standard Transformer acting on the sequence $(x, \Phi(x))$, where $\Phi \in \mathcal{T}_{\text{Rat}}$.

Direction 2 ($\mathcal{F}_{TF} \circ \mathcal{T}_{\text{Rat}} \subseteq \mathcal{F}_{RT}$): Consider an arbitrary composition $F(x) = G(\Phi(x))$, where $G$ is a Transformer and $\Phi$ is a rational function. By definition, $\Phi(x)$ corresponds to the state sequence $\mathbf{h}_t$ of some linear representation. A Rational Transductor can implement this composition by: (1) Configuring its WFA head to generate $\mathbf{h}_t$; (2) Setting the first projection $\mathsf{W}_{\text{proj}}^{(0)}$ to inject $\mathbf{h}_t$ directly into the input embedding (simulating the input to $G$); and (3) Setting subsequent projections $\mathsf{W}_{\text{proj}}^{(l)} = 0$ for $l > 0$. The remaining Transformer layers then implement $G$ exactly. $\square$

**Corollary 25** (Maximality Under Linear Recurrence). *Any extension of finite-depth Transformers that augments the model with a fixed-dimensional, associative, linear state update computable via parallel prefix operations cannot be strictly more expressive than Rational Transductors. Any further expressivity gain requires either non-linear state dynamics or depth growing with input length.*

*Proof.* An "associative, linear state update" is defined by a recurrence of the form $\mathbf{h}_t = \mathsf{M}(x_t)\mathbf{h}_{t-1}$. This is mathematically isomorphic to the state evolution of a Weighted Finite Automaton. Since Transductors are defined (Theorem 24) as the composition of Transformers with the *entire class* of such rational functions $\mathcal{T}_{\text{Rat}}$, they necessarily subsume any specific instance of this recurrence pattern. $\square$

**Minimal Expressive Extension.** Transductors constitute a minimal extension of standard Transformers that suffices to escape the $\text{AC}^0$ barrier while preserving parallelizability. Thus, Transductors represent the smallest algebraically closed class of extensions that enable exact regular-language computation and length generalization.

**Proposition 26** (Virtual Tensorization via Attention). *Let $\mathbf{h}_t \in \mathbb{R}^d$ be the rational feature state at time $t$. The Deep Rational Injection mechanism, combined with a single Self-Attention head, allows the Transductor to compute decision boundaries that are linear in the Kronecker product space $\mathbf{h}_t \otimes \mathbf{h}_{t'}$. Consequently, a Rational Transductor with state dimension $d$ can approximate the expressive power of a higher-order Weighted Finite Automaton with state dimension $d^2$, without explicitly materializing the $O(d^6)$ transition tensor.*

*Proof.* Consider the input to the attention mechanism at layer $l$, denoted by $\widetilde{\mathbf{z}}_t$. Due to Deep Rational Injection, this vector is the sum of the semantic embedding and the projected rational state:

$$\widetilde{\mathbf{z}}_t = \mathbf{z}_t + \mathsf{W}_{\text{proj}}\mathbf{h}_t. \tag{18}$$

The self-attention mechanism computes alignment scores $A_{t,t'}$ between positions $t$ (query) and $t'$ (key) via the inner product of projected representations:

$$\text{Score}(t, t') = (\mathsf{W}_Q\widetilde{\mathbf{z}}_t)^\top(\mathsf{W}_K\widetilde{\mathbf{z}}_{t'}) \tag{19}$$
$$= (\mathbf{z}_t + \mathsf{W}_{\text{proj}}\mathbf{h}_t)^\top\mathsf{W}_Q^\top\mathsf{W}_K(\mathbf{z}_{t'} + \mathsf{W}_{\text{proj}}\mathbf{h}_{t'}). \tag{20}$$

Expanding this quadratic form yields four terms. We focus on the term governing the interaction between the rational states:

$$S_{\text{rational}}(t, t') = \mathbf{h}_t^\top\left(\mathsf{W}_{\text{proj}}^\top\mathsf{W}_Q^\top\mathsf{W}_K\mathsf{W}_{\text{proj}}\right)\mathbf{h}_{t'}. \tag{21}$$

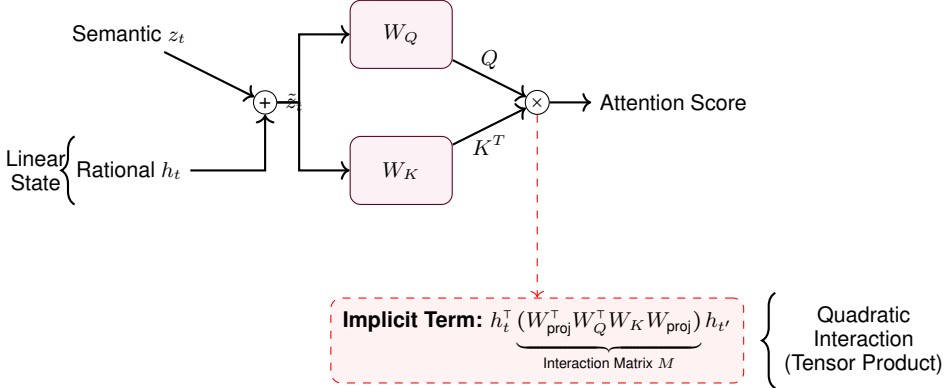

*Figure 9.* Virtual Tensorization. By injecting the linear rational state $h_t$ into the Attention mechanism, the dot product $QK^T$ implicitly computes quadratic terms of the form $h_t^\top M h_{t'}$. This effectively simulates a kernel over the tensor product space $h_t \otimes h_{t'}$, enabling the model to capture higher-order dependencies without explicitly materializing the $O(d^2)$ state space.

Let $M = W_{\text{proj}}^\top W_Q^\top W_K W_{\text{proj}}$ be the effective interaction matrix. Using the vectorization identity $\mathbf{a}^\top M \mathbf{b} = \text{vec}(M)^\top (\mathbf{b} \otimes \mathbf{a})$, we can rewrite the score as:

$$S_{\text{rational}}(t, t') = \text{vec}(M)^\top (\mathbf{h}_{t'} \otimes \mathbf{h}_t). \tag{22}$$

This demonstrates that the attention mechanism implicitly computes a linear function over the tensor product of the states. Physically stacking two WFAs would produce a state space of dimension $d_1 + d_2$, but the attention mechanism allows the model to leverage the multiplicative interactions of the states, effectively simulating a state space of dimension $d \times d$. Thus, the Transductor can capture second-order dependencies (e.g., correlating the state at the start of a clause with the state at the end) that would otherwise require a significantly larger linear automaton to represent. $\square$

**Significance: Implicit State Expansion.** The theoretical significance of this result is twofold. First, it provides a rigorous justification for the "Sidecar" design (wide, parallel recurrence) over the "Stacked" design (deep, serial recurrence) used in recent hybrid models. While stacking linear layers physically increases the state depth, it reintroduces sequential dependencies that hinder training (Gu et al., 2022). In contrast, Virtual Tensorization reveals that the *interaction* between the Rational Head and the Attention mechanism naturally simulates a higher-order automaton. By injecting the linear state $h_t$ into the query/key projections of the attention layer, the dot-product Attention$(Q, K)$ effectively computes a kernel over the tensor product space $h_t \otimes h_{t'}$. This implies that a Rational Transductor with state dimension $d$ can approximate the decision boundaries of a much larger automaton of dimension $d^2$ (or higher, with multiple layers), achieving the expressive benefits of deep recurrence without incurring the $O(L \log T)$ serialization penalty or the $O(d^6)$ cost of explicitly simulating higher-order tensor dynamics. Thus, the architecture achieves a "virtual depth" via the multiplicative interactions of attention, maintaining the optimal $O(\log T)$ parallel efficiency of the linear scan while capturing the complex, hierarchical dependencies usually associated with deep, non-linear stacks.

**Remark: Neural Implementation of Automata Operations.** The Rational Transductor architecture can be understood as a physical realization of the closure properties of Rational Power Series:

- **Parallel Heads as Direct Sum ($\oplus$):** Instantiating multiple independent rational heads (e.g., in the Universal Transductor) corresponds to the **Sum** operation. If head $A$ tracks parity and head $B$ tracks modulo-3, the concatenated state $h = [h_A; h_B]$ represents the direct sum automaton $\mathcal{A}_A \oplus \mathcal{A}_B$, capable of tracking both features in parallel.

- **Attention as Tensor Product ($\otimes$):** The **Intersection** of two regular languages (e.g., "strings that have odd parity AND length $0 \pmod 3$") requires a state space isomorphic to the tensor product of the constituent automata. While the rational layer computes the sum, the subsequent non-linear mixing (via Attention or MLP) approximates the **Cross-Product** or **Hadamard Product**, allowing the model to form complex decision boundaries based on the conjunction of independent rational features.

- **Depth as Composition ($\circ$):** Finally, stacking blocks corresponds to the **Composition** of transductions (or the cascade product), allowing for hierarchical feature extraction where the rational state at layer $l$ depends on the semantic features extracted at layer $l - 1$.

Thus, the architecture provides a complete differentiable substrate for the algebra of rational series.

## C.5. Circuit Complexity Characterization

We now characterize the computational power of Transductors from the perspective of circuit complexity. This framing provides a rigorous explanation for why Transductors solve the Parity Gap.

**Transformers: $\mathsf{AC}^0$, $\mathsf{TC}^0$, and Robustness.** Finite-depth Transformers with hard attention compute Boolean functions within uniform $\mathsf{AC}^0$ (Hahn, 2020). We note that "standard" Transformers using *soft attention* are strictly more expressive, falling into the complexity class $\mathsf{TC}^0$ (Threshold Circuits) (Merrill et al., 2022). While $\mathsf{TC}^0$ models can theoretically express Parity (e.g., via global averaging), they lack the inductive bias to learn these solutions robustly. Recent work identifying C-RASP as a formal model of Transformer length generalization suggests that while tasks like Modulo Counting are in $\mathsf{TC}^0$, they are not expressible in the C-RASP fragment, explaining why standard models fail to learn them in a way that generalizes to unseen lengths. In contrast, Rational Transductors structurally implement the exact $\mathsf{PNC}^1$ mechanism required.

**Theorem 27** (Circuit Upper Bound for Transductors). *For fixed model parameters, any function computed by a Rational Transductor lies in $\mathsf{PNC}^1$ (and strictly in $\mathsf{NC}^1$ under the assumption of finite precision/fields).*

*Proof.* The complexity class $\mathsf{PNC}^1$ (Probabilistic $\mathsf{NC}^1$) characterizes problems solvable by uniform arithmetic circuits of product-depth $O(\log T)$ and polynomial size. The Transductor computation proceeds in two stages:

1. **Rational State Computation.** The rational feature state $\mathbf{h}_t = \mathsf{M}_{x_t} \cdots \mathsf{M}_{x_1} \boldsymbol{\alpha}$ involves an iterated product of $T$ matrices. While iterated multiplication of matrices over a finite field (or bounded width branching programs) is in $\mathsf{NC}^1$ (Barrington, 1989), the simulation of weighted automata over rationals or integers is complete for $\mathsf{PNC}^1$ (Jung, 1985). Since the state update is associative, it admits a parallel-prefix tree decomposition of depth $O(\log T)$, placing it within $\mathsf{PNC}^1$.

2. **Transformer Processing.** Standard Transformer layers perform aggregation (attention) over $T$ elements. Even under the stronger assumption of soft attention (which places Transformers in $\mathsf{TC}^0$), this class is contained within $\mathsf{NC}^1$ (using the standard inclusion $\mathsf{TC}^0 \subseteq \mathsf{NC}^1$) and thus within $\mathsf{PNC}^1$ (Hahn, 2020).

Since $\mathsf{PNC}^1$ is closed under composition and contains the Transformer's complexity class, the total Transductor computation lies in $\mathsf{PNC}^1$. $\qquad\square$

**Remark on Tightness.**  This upper bound is essentially tight from a circuit complexity perspective. The problem of simulating a general WFA (or iterated integer matrix multiplication) is known to be $\mathsf{PNC}^1$-complete. Claiming a tighter bound of $\mathsf{NC}^1$ for the general case would imply $\mathsf{NC}^1 = \mathsf{PNC}^1$, solving a major open problem in complexity theory. However, if we restrict the model to *bounded precision* arithmetic or operations over a finite field (effectively treating the WFA as a DFA or NFA), the complexity collapses to $\mathsf{NC}^1$ via Barrington's Theorem. Rational Transductors thus occupy the complexity class $\mathsf{PNC}^1$, which sits between $\mathsf{NC}^1$ and $\mathsf{L}$ (Log-Space), strictly separating them from the $\mathsf{AC}^0$ (or $\mathsf{TC}^0$) limitations of standard Transformers.

**Proposition 28** (Equivalence to Linear Branching Programs). *The rational feature head of a Transductor implements a linear branching program whose width equals the state dimension $d$. Consequently, Rational Transductors can simulate any bounded-width branching program with polynomial length.*

*Proof.* A Linear Branching Program (LBP) of width $d$ is defined by updates $\mathbf{v}_t = \mathsf{A}(x_t)\mathbf{v}_{t-1}$. This is structurally identical to the WFA recurrence $\mathbf{h}_t = \mathsf{M}_{x_t}\mathbf{h}_{t-1}$. Standard (Boolean) Branching Programs are a subset of LBPs. Since bounded-width permutation branching programs (over a finite alphabet) characterize $\mathsf{NC}^1$ (Barrington's Theorem), and Transductors can learn permutation matrices (as in Theorem 16), this confirms that Transductors possess the expressive power to capture $\mathsf{NC}^1$-complete problems. $\qquad\square$

**Theorem 29** (Circuit Lower Bound for Transductors). *There exist functions computable by Rational Transductors that are not in $\mathsf{AC}^0$.*

*Proof.* By Theorem 16 and Theorem 17, Transductors can uniformly compute Parity and Modular Counting over unbounded input lengths. Since $MOD_k$ is not computable in $\mathsf{AC}^0$ for any $k \geq 2$ (Smolensky, 1987), this establishes that $\mathcal{F}_{\mathrm{RT}} \nsubseteq \mathsf{AC}^0$. $\qquad\square$

**Corollary 30** (Circuit Complexity Sandwich)**.** *The class of functions computable by Rational Transductors satisfies the following inclusion hierarchy relative to standard (hard-attention) Transformers:*

$$\mathsf{AC}^0 \subsetneq \mathcal{F}_{\mathrm{RT}} \subseteq \mathsf{PNC}^1.$$

**The Parallelism Gap** This hierarchy highlights the computational trade-off characterizing Rational Transductors. While strictly more expressive than standard hard-attention Transformers ($\mathsf{AC}^0$), Transductors remain within $\mathsf{PNC}^1$, ensuring $O(\log T)$ parallelizability. This strictly separates them from general non-linear RNNs (like LSTMs or GRUs with unbounded precision), which are P-complete under sequential updates and cannot be parallelized to logarithmic depth without approximation.

**Uniformity Across Lengths.** All expressivity results for Transductors are *uniform*: a single fixed set of parameters suffices to compute the target function for sequences of arbitrary length. No depth or parameter scaling is required. In particular, Transductors correspond to uniform circuit families rather than length-specific (non-uniform) constructions.

**On Real-Valued Attention.** All circuit complexity claims refer to the standard Boolean abstraction of arithmetic operations with bounded precision, following prior analyses of Transformers in $\mathsf{AC}^0$ and $\mathsf{NC}^1$.

**Why $\mathsf{PNC}^1$ Is Essentially Tight.** This upper bound is not an artifact of our proof technique but a reflection of the intrinsic hardness of the rational state update. The rational state component corresponds to integer matrix multiplication or weighted automata evaluation, problems known to be complete for $\mathsf{PNC}^1$ (Jung, 1985). Since Transductors do not permit non-associative state updates, they cannot simulate $\mathsf{NC}^2$-hard problems such as iterated matrix powering with growing dimension, nor can they solve L-complete problems like graph connectivity (unless $\mathsf{PNC}^1 = \mathsf{L}$). Thus, the architecture is precisely situated: strictly more expressive than $\mathsf{AC}^0$ but bounded by the parallel limits of associative recurrence.

# D. Theoretical Analysis of Learning

While the architecture of Rational Transductors is linear and convex in parts, its training dynamics in the learnable regime require careful analysis. In this section, we analyze the theoretical properties of the hypothesis class, distinguishing between the universality of random features at initialization and the optimization stability of learned features.

## D.1. Universality and Efficiency of Random Features

We first address whether a randomly initialized WFA can serve as a generic state encoder. We prove that a sufficiently large random WFA generates a state space rich enough to linearly reconstruct the state of *any* target WFA up to a finite horizon, justifying our use of near-identity initialization strategies.

**Theorem 31** (Universality of Random Rational Features). *Let $\mathcal{A}^*$ be a target WFA with state dimension $d^*$ generating states $\mathbf{h}_t^* \in \mathbb{R}^{d^*}$. Let $\mathcal{X}_L$ be the set of all input sequences of length up to $L$. Let $\mathcal{A}_{rand}$ be a random WFA with state dimension $d \geq |\mathcal{X}_L|$, initialized such that its transition matrices $\{\mathsf{M}_\sigma\}$ are drawn from a continuous distribution (e.g., Gaussian). With probability one (assuming generic initialization and no algebraic dependencies), there exists a linear projection matrix $\mathsf{W} \in \mathbb{R}^{d^* \times d}$ such that for all sequences $x \in \mathcal{X}_L$, the random feature state $\mathbf{h}_t$ perfectly recovers the target state:*

$$\mathsf{W}\mathbf{h}_t = \mathbf{h}_t^* \tag{23}$$

*Proof.* The proof relies on showing that the random WFA maps distinct histories to linearly independent vectors, thereby forming a basis for the target dynamics.

1. Matrix Formulation of Trajectories. Let $N = |\mathcal{X}_L|$ be the number of distinct prefixes of length $\leq L$. Let $\mathbf{H}^* \in \mathbb{R}^{d^* \times N}$ be the matrix collecting the target states $\mathbf{h}_t^*$ for all $N$ prefixes as columns. Let $\mathbf{H} \in \mathbb{R}^{d \times N}$ be the matrix collecting the random WFA states $\mathbf{h}_t$ for the same prefixes. Our goal is to find $\mathsf{W}$ satisfying $\mathsf{W}\mathbf{H} = \mathbf{H}^*$. This system of linear equations has an exact solution if and only if the row space of $\mathbf{H}^*$ is contained in the row space of $\mathbf{H}$. A sufficient condition is that $\mathbf{H}$ has full column rank (rank $N$).

2. Linear Independence of Random States. The state $\mathbf{h}_t$ for a sequence $x = (x_1, \ldots, x_t)$ in the random WFA is given by the product $\mathbf{h}_t = \mathsf{M}_{x_t} \ldots \mathsf{M}_{x_1} \boldsymbol{\alpha}$. This is a polynomial map in the entries of the matrices $\{\mathsf{M}_\sigma\}$. Consider the determinant of any $N \times N$ submatrix of $\mathbf{H}$. This determinant is a polynomial function of the random weights. Since we can construct at least one specific instance of WFA parameters where distinct sequences map to linearly independent vectors (e.g., by mapping to distinct basis vectors), this polynomial is not identically zero. A fundamental result in algebra states that if a polynomial is not identically zero, its zero set has measure zero. Since the entries of $\{\mathsf{M}_\sigma\}$ are drawn from a continuous distribution, the vectors in $\mathbf{H}$ are linearly independent with probability 1, provided $d \geq N$.

3. Existence of Projection. Since $\mathbf{H}$ has rank $N$ (full column rank), its Moore-Penrose pseudoinverse $\mathbf{H}^+ = (\mathbf{H}^\top \mathbf{H})^{-1} \mathbf{H}^\top$ is a well-defined left inverse (satisfying $\mathbf{H}^+\mathbf{H} = \mathsf{I}$). We construct the projection as:

$$\mathsf{W} = \mathbf{H}^* \mathbf{H}^+. \tag{24}$$

Verifying the reconstruction:

$$\mathsf{W}\mathbf{H} = \mathbf{H}^*(\mathbf{H}^+\mathbf{H}) = \mathbf{H}^*\mathsf{I} = \mathbf{H}^*. \tag{25}$$

Thus, the random features $\mathbf{h}_t$ contain sufficient information to linearly reconstruct the target states $\mathbf{h}_t^*$ exactly. $\square$

**Approximation Bounds.** While Theorem 31 establishes that random rational features can represent any target state space given sufficient width, we must also quantify how the approximation error scales with the dimension $d$. This is critical for understanding the efficiency of the random initialization. We identify the infinite-width limit of the Rational Transductor as an implicit kernel, which we term the *induced rational kernel*.

**Theorem 32** (Uniform Spectral Approximation Bound). *Let $\mathcal{M} = \{\mathsf{M}_\sigma\}_{\sigma \in \Sigma}$ be the collection of transition matrices. Let $\mathcal{D}$ be a distribution over $\mathbb{R}^{d \times d}$ used to initialize each $\mathsf{M}_\sigma$ independently. Consider the kernel $K : \Sigma^* \times \Sigma^* \to \mathbb{R}$ defined by the expectation over this random initialization:*

$$K(x, x') = \underset{\mathcal{M} \sim \mathcal{D}}{\mathbb{E}}[\langle \mathsf{M}_x \boldsymbol{\alpha}, \mathsf{M}_{x'} \boldsymbol{\alpha} \rangle], \tag{26}$$

*where $\mathsf{M}_x = \mathsf{M}_{x_t} \ldots \mathsf{M}_{x_1}$. Let $f^*$ be a target function (the true sequence labeling function we wish to learn) that lies in the Reproducing Kernel Hilbert Space (RKHS) $\mathcal{H}_K$ with norm $\|f^*\|_K \leq C$.*

*Let $f_d(x)$ be the function computed by a Rational Transducer with $d$ random features, where the transition matrices are drawn from $\mathcal{D}$ and fixed, and only the linear readout $\mathbf{w}$ is trained:*

$$f_d(x) = \mathbf{w}^\top (\mathsf{M}_x \alpha). \tag{27}$$

*Assume the input domain $\mathcal{X}$ is restricted to sequences of length at most $T_{\max}$ equipped with a weighted Hamming (or edit) metric, and the initialization distribution $\mathcal{D}$ satisfies the spectral constraint $\sup_\sigma \|\mathsf{M}_\sigma\|_2 \le 1$ almost surely. For any $\delta > 0$, with probability at least $1 - \delta$ over the random draw of $\mathcal{M}$, the uniform approximation error is bounded by:*

$$\sup_{x \in \mathcal{X}} |f^*(x) - f_d(x)| \le \frac{C}{\sqrt{d}} \left( \sqrt{2 \log \frac{1}{\delta}} + 4\sqrt{\frac{d}{d-1} \Delta L} \right), \tag{28}$$

*where $\Delta$ is the diameter of the metric space $\mathcal{X}$ and $L$ is the Lipschitz constant of the rational feature map $x \mapsto \mathsf{M}_x \alpha$ induced by the spectral bounds.*

**Remark on the RKHS Assumption (Realizability).** The condition $\|f^*\|_K \le C$ is formally equivalent to an assumption of realizability. By definition, the RKHS $\mathbb{H}_K$ consists of all functions that can be expressed as limits of linear combinations of the random rational features defined by the distribution $\mathcal{D}$. Since Theorem 31 establishes that these random features form a universal basis for the state dynamics of *any* Weighted Finite Automaton (up to a given length), the RKHS effectively covers the entire class of Rational Power Series. Thus, assuming $f^*$ lies in this space is simply assuming that the ground truth is, in fact, a regular language (or rational series) that the Rational Transducer architecture is capable of representing.

*Proof.* The proof proceeds by establishing boundedness and continuity of the random features, then applying uniform concentration bounds.

1. Boundedness via contraction. By the stability-aware parameterization (Section 3.1), the transition matrices are spectrally normalized such that $\|\mathsf{M}_\sigma\|_2 \le \gamma \le 1$. Consequently, for any sequence $x$ of length $T$, the feature vector $h(x) = \mathsf{M}_{x_T} \dots \mathsf{M}_{x_1} \alpha$ satisfies:

$$\|h(x)\|_2 \le \left[ \prod_{t=1}^{T} \|\mathsf{M}_{x_t}\|_2 \right] \|\alpha\|_2 \le \gamma^T \|\alpha\|_2 \le \|\alpha\|_2. \tag{29}$$

Thus, the random features $\phi(x) = h(x)$ are uniformly bounded in Euclidean norm by $R = \|\alpha\|_2$ independent of sequence length. This ensures that the kernel $K(x, x')$ is bounded, satisfying the preconditions for Hoeffding-type concentration.

2. Lipschitz continuity. We equip $\Sigma^{\le T_{\max}}$ with a weighted Hamming metric under which $x \mapsto \mathsf{M}_x \alpha$ is $L$-Lipschitz due to spectral contraction. Since the map $x \mapsto \mathsf{M}_x \alpha$ is a composition of contractive linear maps, it is Lipschitz continuous with respect to the initialization parameters. Specifically, a perturbation in the input (interpreted as a perturbation in the effective operator sequence) results in a bounded deviation in the output state $h(x)$, with Lipschitz constant $L$ determined by the spectral bounds of $\mathcal{M}$.

3. Uniform convergence. Let $f_d(x) = \langle w, \phi(x) \rangle$ be the random feature approximation. The error $\Gamma(x) = f^*(x) - f_d(x)$ is a sum of $d$ independent, bounded random variables.

- Pointwise Bound: For a fixed $x$, Hoeffding's inequality gives

$$\mathbb{P}(|\Gamma(x)| > \epsilon) \le 2 \exp(-d\epsilon^2 / 2C^2 R^2).$$

- Covering Argument: Let $\mathcal{N}(\epsilon, \mathcal{X})$ be the $\epsilon$-covering number of the domain. Since $\Gamma(x)$ is Lipschitz with constant $L' \propto CL$, we can approximate the supremum over $\mathcal{X}$ by the maximum over the cover centers. Applying the union bound over the cover and optimizing the scale $\epsilon$ (via chaining) yields the additional term proportional to $\sqrt{\Delta L}$.

Combining these yields the stated bound, which holds with high probability over the random initialization. $\square$

**Interpretation.** This result fundamentally reframes the challenge of learning sequential logic. The bound $O(C/\sqrt{d})$ indicates that the approximation error is governed solely by the *scale* of the random projection $d$ and the *complexity* of the target function $C$ (in the RKHS sense). This implies a direct resource trade-off: we can avoid the optimization instability

of training deep recurrences (the "non-convex hard problem") by simply increasing the width $d$ of the random state (the "linear scaling problem"). Crucially, as $d \to \infty$, the random rational features form a universal basis for the class of functions defined by the rational kernel, guaranteeing that a sufficiently wide Rational Transductor can solve any task representable by stable linear dynamics without ever training the recurrent weights.

**Connection to Rational Kernels.** The kernel $K$ defined above is a specific instance of a *Rational Kernel* as defined by Cortes et al. (2004). Specifically, if the transition matrices are drawn such that $\mathbb{E}[\mathsf{M}_\sigma \otimes \mathsf{M}_\sigma]$ is stable, $K(x, x')$ corresponds to the rational series computed by a weighted automaton on the product monoid. Our Random Rational Features can thus be viewed as an efficient, randomized approximation explicitly designed to scale these classical kernels to deep learning contexts.

**Geometry Preservation via Johnson-Lindenstrauss**. The $O(1/\sqrt{d})$ scaling in Theorem 32 acts as a functional equivalent to the Johnson-Lindenstrauss (JL) Lemma for sequence histories. Conceptually, the true history of the sequence lives in an infinite-dimensional feature space $\mathcal{H}_K$ induced by the rational kernel. The random recurrent head acts as a random projection operator $P : \mathcal{H}_K \to \mathbb{R}^d$. By the JL lemma, pairwise distances (and thus the distinguishability of distinct histories) are preserved with distortion $\epsilon$ provided $d = \Omega(\epsilon^{-2} \log N)$. This explains the "inefficiency" highlighted in Proposition 33: random features rely on generic concentration of measure, whereas learned features exploit the specific algebraic structure of the task.

**The Compactness Gap.** While Theorem 31 shows representational completeness, it relies on an exponentially large state dimension $d \approx |\Sigma|^L$. In contrast, learning the transitions allows for compact representations. We quantify this gap below.

**Proposition 33** (Compactness Gap). *Let the target language be the set of strings with an even number of 1s (Parity).*

- ***Learned Regime:*** *A Rational Transductor can solve this perfectly with state dimension $d = 2$ by learning the exact flip transition $\mathsf{M}_1 = \begin{pmatrix} 0 & 1 \\ 1 & 0 \end{pmatrix}$.*

- ***Random Regime:*** *To solve this with fixed random features and a linear readout with error $\epsilon < 1/2$, the required dimension scales as $d = \Omega(1/\epsilon^2)$ (by Theorem 32). (This lower bound holds for any isotropic initialization distribution.)*

*Proof.* Part 1: Learned Features (Exact Construction). We explicitly construct a WFA with $d = 2$ that computes parity. Let the state space basis correspond to states $\{\text{Even}, \text{Odd}\}$. Initialize $h_0 = (1, 0)^\top$. Define the transition matrices for input bits $0$ and $1$ as:

$$\mathsf{M}_0 = \begin{pmatrix} 1 & 0 \\ 0 & 1 \end{pmatrix} = I, \quad \mathsf{M}_1 = \begin{pmatrix} 0 & 1 \\ 1 & 0 \end{pmatrix} = \sigma_x. \tag{30}$$

For any sequence $x$, the state $h_T = \mathsf{M}_{x_T} \dots \mathsf{M}_{x_1} h_0$ will be $(1, 0)^\top$ if the number of ones is even, and $(0, 1)^\top$ if odd. A linear readout $y = w^\top h_T$ with $w = (-1, 1)^\top$ yields $y = -1$ (Even) or $y = 1$ (Odd). The separation is perfect (margin = 2), so the error is zero with $d = 2$.

Part 2: Random Features (Statistical Lower Bound). In the Random Regime, the function approximation $f_d(x)$ is a sum of $d$ i.i.d. random features. By the Central Limit Theorem, the variance of the estimator scales as $\text{Var}(f_d(x)) \propto \frac{1}{d}$. To correctly classify parity for all inputs with high probability, the approximation error (noise) must be strictly less than the classification margin (signal). From Theorem 32 (see above), the worst-case error scales as $O(1/\sqrt{d})$. Specifically, to ensure $|f^*(x) - f_d(x)| < \epsilon$ uniformly, we require the standard deviation $\sigma_d \approx \frac{1}{\sqrt{d}}$ to be suppressed below $\epsilon$. Rearranging $\frac{1}{\sqrt{d}} \le \epsilon$ yields $d \ge \frac{1}{\epsilon^2}$. Thus, achieving a fixed error tolerance $\epsilon$ requires the dimension to scale quadratically with the inverse error, whereas the learned solution achieves zero error with constant dimension. $\square$

The *random regime* results of this subsection primarily serve to justify our *near-identity initialization* (detailed in Appendix E.3).

**Corollary 34** (Convex Learnability of Rational Projections). *Let $\mathcal{L}(\cdot, \cdot)$ be a convex loss function (e.g., squared error or cross-entropy). Consider a Rational Transductor where the WFA transitions $\mathcal{M}$ are fixed and the downstream Transformer parameters $\theta$ are fixed (or constitute a linear readout). The optimization problem for the projection matrix $\mathsf{W}$*

$$\min_{\mathsf{W}} \sum_{(x,y) \in \mathcal{D}} \mathcal{L}(y, G_\theta(x) + \mathsf{W}\mathbf{h}_T(x)). \tag{31}$$

*is a convex optimization problem.*

*Proof.* Since the WFA transitions are fixed, the feature vector $\mathbf{h}_T(x)$ is a constant vector for any given input $x$. The term $\mathrm{W}\mathbf{h}_T(x)$ is linear in the variable W. Consequently, the model output $\widehat{y} = G_\theta(x) + \mathrm{W}\mathbf{h}_T(x)$ is an affine function of W. A fundamental property of convex optimization is that the composition of a convex function with an affine mapping preserves convexity. Therefore, the objective function, being a sum of convex functions composed with affine maps, is convex with respect to W. This guarantees that gradient descent will converge to a global optimum. $\square$

### D.2. Optimization Dynamics

In the learned regime, we must ensure that the optimization landscape is well-behaved. We provide three key theorems guaranteeing optimization stability and well-conditioned gradient flow.

**Theorem 35** (Gradient Norm Preservation). *Consider the gradient of the loss $\mathcal{L}$ with respect to the hidden state $\mathbf{h}_t$, denoted $\boldsymbol{\delta}_t = \nabla_{h_t}\mathcal{L}$. In the backward pass, the error signal propagates as*

$$\boldsymbol{\delta}_{t-1} = \mathrm{M}_{x_t}^\top \boldsymbol{\delta}_t + \mathbf{v}_{t-1}. \tag{32}$$

1. **Explosion Guarantee:** *If the transitions satisfy the spectral constraint $\|\mathrm{M}_\sigma\|_2 \leq \gamma \leq 1$, then the propagated gradient norm grows at most linearly (or is bounded if $\gamma < 1$): $\|\boldsymbol{\delta}_{t-k}\|_2 \leq \gamma^k \|\boldsymbol{\delta}_t\|_2 + C$, preventing exponential explosion.*

2. **Vanishing Guarantee:** *If the transitions are parameterized to be orthogonal ($\mathrm{M}_\sigma^\top \mathrm{M}_\sigma = I$), then the gradient norm is exactly preserved in the absence of injection: $\|\boldsymbol{\delta}_{t-k}\|_2 = \|\boldsymbol{\delta}_t\|_2$.*

*Proof.* This follows directly from the properties of the spectral norm under matrix multiplication. For (1), $\|\boldsymbol{\delta}_{t-1}\| \leq \|\mathrm{M}_{x_t}^\top\|\|\boldsymbol{\delta}_t\| + \|v\| \leq \gamma\|\boldsymbol{\delta}_t\| + \|v\|$. By induction, the homogeneous component decays (or stays constant) while the additive injection terms accumulate linearly, preventing exponential growth. For (2), if $\mathrm{M}_{x_t}$ is orthogonal, $\|\mathrm{M}_{x_t}^\top \boldsymbol{\delta}_t\|_2 = \|\boldsymbol{\delta}_t\|_2$, ensuring the error signal from time $t$ reaches time $0$ without attenuation. This effectively solves the "long-term dependency" problem for the linear component. $\square$

This result establishes a structural immunity to the *gradient explosion problem*. Unlike standard RNNs, which often undergo chaotic gradient growth and require heuristic fixes like clipping, Rational Transductors are mathematically guaranteed to remain stable. By strictly bounding the spectral radius $\gamma \leq 1$, the architecture ensures that error signals never expand exponentially, regardless of sequence length.

**Theorem 36** (Gradient Maintenance via Deep Injection). *Consider the backward recurrence for the Rational Transductor error signal (Eq. 32): $\boldsymbol{\delta}_{t-1} = M^\top \boldsymbol{\delta}_t + \mathbf{v}_{t-1}$. Let the transition matrix be contractive with $\|M\|_2 \leq \gamma < 1$. Let $\mathbf{v}_{t-1}^{(l)} = \nabla_{h_{t-1}}\mathcal{L}^{(l)}$ denote the gradient contribution from the injection at layer $l$. The gradient norm at step $t - k$ is bounded below by the immediate injection terms:*

$$\|\boldsymbol{\delta}_{t-k}\|_2 \geq \left\|\sum_{j=0}^{k-1} (\mathrm{M}^\top)^j \mathbf{v}_{t-1-j}\right\|_2 - \gamma^k \|\boldsymbol{\delta}_t\|_2 \tag{33}$$

*Proof.* Unrolling the recurrence $\boldsymbol{\delta}_{t-1} = \mathrm{M}^\top \boldsymbol{\delta}_t + \mathbf{v}_{t-1}$ for $k$ steps yields:

$$\boldsymbol{\delta}_{t-k} = (\mathrm{M}^\top)^k \boldsymbol{\delta}_t + \sum_{j=0}^{k-1} (\mathrm{M}^\top)^j \mathbf{v}_{t-1-j}, \tag{34}$$

where $(\mathrm{M}^\top)^k$ denotes the ordered product of transition matrices. Applying the reverse triangle inequality ($\|\mathbf{a}+\mathbf{b}\| \geq \|\mathbf{b}\| - \|\mathbf{a}\|$) and the sub-multiplicative property of the spectral norm ($\|(\mathrm{M}^\top)^k \boldsymbol{\delta}_t\| \leq \gamma^k \|\boldsymbol{\delta}_t\|$), we obtain the lower bound:

$$\|\boldsymbol{\delta}_{t-k}\|_2 \geq \left\|\sum_{j=0}^{k-1} (\mathrm{M}^\top)^j \mathbf{v}_{t-1-j}\right\|_2 - \gamma^k \|\boldsymbol{\delta}_t\|_2. \tag{35}$$

The first term represents the accumulated gradient injections from the deep Transformer layers. Even if the temporal connection vanishes (i.e., the second term $\gamma^k \|\boldsymbol{\delta}_t\| \to 0$ as $k \to \infty$), the state $\mathbf{h}_{t-k}$ retains the magnitude of the direct gradient signal $\mathbf{v}$, preventing total gradient collapse. $\square$

Crucially, this theorem demonstrates how Deep Rational Injection mitigates the *vanishing gradient problem*. In traditional recurrences, gradients depend entirely on backpropagation through time and often decay to zero. Here, the injection mechanism creates *gradient highways* that provide direct supervision from the local Transformer layers to the recurrent state. This allows the model to learn representations for recent events even if long-term dependencies are temporarily weak.

**Theorem 37** (Bounded Hessian and Smoothness). *Let $\mathcal{L}(M)$ be the loss with respect to the transition matrix $M$, assuming a contractive spectral constraint $\|M\|_2 \le \gamma < 1$ and a Lipschitz-smooth downstream Transformer (e.g., with LayerNorm and bounded activation derivatives). Then, the loss function is $\beta$-smooth with respect to $M$. That is, the spectral norm of the Hessian is uniformly bounded independent of sequence length:*

$$\left\|\nabla_M^2 \mathcal{L}\right\|_2 \le \beta(\gamma, C) < +\infty. \tag{36}$$

*Proof.* The hidden state $h_t$ is a polynomial in M of degree $t$. The second derivative $\frac{\partial^2 \mathbf{h}_t}{\partial M^2}$ involves terms of the form $\sum_{i,j} \mathsf{M}^i(\partial \mathsf{M})\mathsf{M}^j(\partial \mathsf{M})\mathsf{M}^{t-i-j-2}$. For a standard RNN without constraints, these terms sum to a magnitude scaling with $t^2\|\mathsf{M}\|^{t-2}$, which explodes if $\|\mathsf{M}\| > 1$. However, under the strict contraction constraint $\|\mathsf{M}\| \le \gamma < 1$, the series of second derivative terms converges geometrically. Specifically, the sum is bounded by the second derivative of the geometric series $(1-\gamma)^{-1}$, which is $2(1-\gamma)^{-3}$. Since the Hessian of the composition $\mathcal{L}(\mathbf{h}_t(\mathsf{M}))$ depends on $\nabla \mathbf{h}_t$ and $\nabla^2 \mathbf{h}_t$ (both bounded by geometric series) and the smooth Transformer readout, the total Hessian is uniformly bounded. $\square$

Practically, the boundedness of the Hessian ensures a *well-conditioned optimization landscape*. This implies that the loss surface is free of pathological curvature or sharp cliffs, allowing standard first-order optimizers (like AdamW) to navigate the parameter space efficiently without requiring complex second-order corrections.

### D.3. Generalization and Robustness

We now examine the model's ability to generalize to unseen lengths and withstand adversarial perturbations.

**Theorem 38** (Time-Invariant Error Bounding). *Let $M^*$ be the true transition logic of the target task (e.g., a counter or automaton) and let $\widehat{\mathsf{M}}$ be the learned transition matrix. Assume the learned dynamics are contractive with spectral norm $\|\widehat{\mathsf{M}}\|_2 \le \gamma < 1$. If the learned matrix approximates the true logic with error $\|\widehat{\mathsf{M}} - M^*\| \le \epsilon$, then the deviation between the true state $h_t^*$ and the Rational Transductor state $\widetilde{\mathbf{h}}_t$ is uniformly bounded for all $t > 0$:*

$$\sup_{t \ge 1} \left\|\mathbf{h}_t^* - \widetilde{\mathbf{h}}_t\right\| \le \frac{\epsilon C}{1 - \gamma}, \tag{37}$$

*where $C = \sup_t \|\mathbf{h}_{t-1}^*\|$ is the bound on the true state magnitude.*

*Proof.* Let $e_t = h_t^* - \widehat{h}_t$ be the state error at time $t$. The evolution of the error is given by:

$$\mathbf{e}_t = M^*\mathbf{h}_{t-1}^* - \widehat{\mathsf{M}}\widetilde{\mathbf{h}}_{t-1} \tag{38}$$

$$= M^*\mathbf{h}_{t-1}^* - (M^* + \Delta)(\mathbf{h}_{t-1}^* - \mathbf{e}_{t-1}) \quad \text{where } \Delta = \widehat{\mathsf{M}} - M^* \tag{39}$$

$$= \widehat{\mathsf{M}}\mathbf{e}_{t-1} - \Delta\mathbf{h}_{t-1}^*. \tag{40}$$

Taking norms and applying the triangle inequality:

$$\|\mathbf{e}_t\| \le \|\widehat{\mathsf{M}}\|\|\mathbf{e}_{t-1}\| + \|\Delta\|\|\mathbf{h}_{t-1}^*\| \le \gamma\|\mathbf{e}_{t-1}\| + \epsilon C. \tag{41}$$

This is a linear recurrence inequality. Solving for the steady state (limit as $t \to \infty$) yields the geometric series sum:

$$\|\mathbf{e}_t\| \le \epsilon C \sum_{k=0}^{t-1} \gamma^k \le \frac{\epsilon C}{1 - \gamma}. \tag{42}$$

This completes the proof. $\square$

**Remark: The Unitary Regime** ($\gamma = 1$). Theorem 38 relies on the contraction coefficient $\gamma < 1$ to bound the propagation of error, characteristic of the "fading memory" regime required for stable approximate matching. For tasks like Parity or Modular Counting which require infinite memory ($\gamma = 1$), generalization is instead guaranteed by the *algebraic exactness* of the orthogonal parameterization (Theorem 17). In this unitary regime, the transition matrices $M_\sigma$ form a subgroup isomorphic to the target automaton, ensuring that the state evolution remains exactly on the solution manifold regardless of sequence length $T$, provided the precision is sufficient.

**Implication: Learned Positional Encodings.** This result provides the theoretical justification for disabling explicit Positional Encodings (PEs) in algorithmic tasks. Standard PEs (RoPE, sinusoidal) are brittle because they introduce out-of-distribution drift when test sequences exceed training lengths ($L_{\text{test}} > L_{\text{train}}$). By relying solely on the recurrent state update, Rational Transductor learn a *time-invariant* transition rule. Theorem 38 guarantees that the error of this rule does not compound over time but remains bounded by a constant, unlocking the perfect length generalization observed experiments (Section 6.2).

**Theorem 39** (Length-Independent Generalization). *Let $\mathcal{F}_{RT}$ be the class of Rational Feature functions $f(x) = \mathbf{w}^\top(M_{x_T} \ldots M_{x_1} \boldsymbol{\alpha})$ parametrized by transition matrices satisfying the spectral constraint $\|M_\sigma\|_2 \leq \gamma < 1$ and bounded readout $\|\mathbf{w}\|_2 \leq W$. For a dataset $S$ of $N$ sequences of length $T$, the Empirical Rademacher Complexity is bounded by:*

$$\widehat{\mathfrak{R}}_S(\mathcal{F}_{RT}) \leq \frac{W\|\boldsymbol{\alpha}\|_2}{\sqrt{N}}\left(\frac{1}{1-\gamma}\right). \tag{43}$$

*Proof.* The output of the Transductor head at time $T$ can be written recursively. Since the transition dynamics are linear and contractive, the sensitivity of the output $h_T$ to the input token at position $t$ decays exponentially as $\gamma^{T-t}$. Following standard stability analysis for recurrent systems (Miller & Hardt, 2018), the Lipschitz constant of the map $x \to h_T$ with respect to the sequence (in the $\ell_2$ sense) is bounded by the geometric series $\sum_{k=0}^{T} \gamma^k < \frac{1}{1-\gamma}$. By Talagrand's contraction lemma (Ledoux & Talagrand, 1991), the complexity of the class is bounded by the product of this Lipschitz constant and the complexity of the input embedding layer (which is $O(1/\sqrt{N})$). Crucially, because the geometric series converges, the bound is independent of $T$. $\qquad\square$

**Interpretation: Infinite-Horizon Reliability.** Standard generalization bounds for RNNs typically scale with the sequence length (e.g., $O(T/\sqrt{N})$ or $O(\sqrt{T/N})$), implying that model performance degrades on longer tasks. Theorem 39 establishes that for contractive Rational Transductors, the complexity—and thus the generalization gap—is *independent of sequence length $T$*. This theoretically guarantees that the model can be deployed on streaming data of indefinite duration without the risk of overfitting to the specific length statistics of the training set.

**Theorem 40** (Hankel-Rademacher Complexity (Balle & Mohri, 2017)). *Let $\mathcal{F}_{Hankel,r}$ be the class of rational functions $f : \Sigma^* \to \mathbb{R}$ with Hankel nuclear norm $\|H_f\|_{\mathcal{S}_1} \leq r$. Let $S = (x_1, \ldots, x_N)$ be a sample of sequences and let $L_{\max} = \max_{x \in S} |x|$ be the maximum sequence length. Define the* max-prefix collision *term $\sigma_S^2 = \sup_{u \in \Sigma^*} \sum_{i=1}^{N} \mathbb{I}[u \in pref(x_i)]$. The Empirical Rademacher complexity is bounded by:*

$$\widehat{\mathfrak{R}}_S(\mathcal{F}_{Hankel,r}) \leq \frac{r}{N}\left(\sqrt{2\sigma_S^2 \log(2D_S)} + \frac{2}{3}\log(2D_S)\right) = \widetilde{O}\left(\frac{r}{\sqrt{N}}\right), \tag{44}$$

*where $D_S$ is the total number of distinct suffixes present in the sample $S$ (the size of the suffix trie), which is bounded by $NL_{\max}$.*

*Proof.* This result is a direct application of Theorem 6 from Balle & Mohri (2017). The proof relies on two key steps:

1. **Fliess' Theorem Duality:** The condition $\|H_f\|_{\mathcal{S}_1} \leq r$ on the Hankel matrix is dual to the spectral norm bound on the data matrix $\mathbf{Y}$ (defined over the prefix/suffix splits of the sample $S$).

2. **Matrix Concentration:** The expected spectral norm of the data matrix is bounded using the Matrix Bernstein inequality (specifically, a non-commutative Khintchine inequality). The variance term in this inequality corresponds exactly to the maximum number of times any specific prefix $u$ appears across the dataset sequences $x_i$, denoted by $\sigma_S^2$.

For a detailed derivation of the constants 2/3 and the logarithmic factor, we refer the reader to the original proof in Balle & Mohri (2017). $\qquad\square$

**Significance: Low-Rank Regularization.** This result is profound because the rank of the Hankel matrix corresponds exactly to the minimal state dimension $d_{\min}$ of the Weighted Finite Automaton (Fliess' Theorem). Consequently, penalizing the nuclear norm of the Hankel matrix (which our "Diagonal + Low Rank" parameterization effectively does) is rigorously equivalent to regularizing the *state dimension* of the underlying latent automaton. This confirms that Rational Transductors generalize by learning low-rank algebraic structures, distinguishing them from standard Transformers which often overfit to high-rank, spurious correlations.

**Theorem 41** (Lipschitz Input Stability). *Let $x = (\mathbf{x}_1, \ldots, \mathbf{x}_T)$ be a sequence of continuous input embeddings where $\mathbf{x}_t \in \mathbb{R}^{d_{in}}$. Let $f_{RT}(x)$ be the output of the rational head. Assume the transition dynamics are contractive ($\|\mathsf{M}(\mathbf{x})\|_2 \le \gamma < 1$) and the encoding of inputs into matrices is Lipschitz continuous with constant $K_M$ (i.e., $\|\mathsf{M}(\mathbf{x}) - \mathsf{M}(\mathbf{x}')\| \le K_M \|\mathbf{x} - \mathbf{x}'\|$). Then, the map from the input sequence $x$ to the final state $\mathbf{h}_T$ is Lipschitz continuous with constant:*

$$L_{seq} \le \frac{K_M \|\boldsymbol{\alpha}\|}{1 - \gamma} \tag{45}$$

*This bound is independent of the sequence length $T$.*

*Proof.* Let $x = (\mathbf{x}_1, \ldots, \mathbf{x}_T)$ and $x' = (\mathbf{x}'_1, \ldots, \mathbf{x}'_T)$ be two input sequences differing at time step $t$. Let $\mathbf{h}_k$ and $\mathbf{h}'_k$ be the respective state sequences. The state deviation $\mathbf{e}_k = \mathbf{h}_k - \mathbf{h}'_k$ evolves as:

$$\mathbf{e}_k = \mathsf{M}(\mathbf{x}_k)\mathbf{e}_{k-1} + (\mathsf{M}(\mathbf{x}_k) - \mathsf{M}(\mathbf{x}'_k))\mathbf{h}'_{k-1} \tag{46}$$

Taking norms:

$$\|\mathbf{e}_k\| \le \gamma \|\mathbf{e}_{k-1}\| + K_M \|\mathbf{x}_k - \mathbf{x}'_k\| R, \tag{47}$$

where $R = \sup \|\mathbf{h}'_k\|$ is the bounded state norm. Iterating this recurrence, the total deviation at time $T$ due to a perturbation at time $t$ is bounded by $K_M R \gamma^{T-t} \|\mathbf{x}_t - \mathbf{x}'_t\|$. Summing over all possible perturbation times $t$ (triangle inequality for the whole sequence norm):

$$\|\mathbf{h}_T - \mathbf{h}'_T\| \le \sum_{t=1}^{T} K_M R \gamma^{T-t} \|\mathbf{x}_t - \mathbf{x}'_t\| \le \left( K_M R \sum_{k=0}^{\infty} \gamma^k \right) \sup_t \|\mathbf{x}_t - \mathbf{x}'_t\|. \tag{48}$$

The geometric series converges to $(1 - \gamma)^{-1}$, yielding the length-independent bound. $\square$

**Interpretation: Robustness to Embedding Noise.** This theorem guarantees that Rational Transductors are robust to small perturbations in the input embeddings, such as those caused by quantization, noise, or minor distribution shifts. The state $\mathbf{h}_T$ varies smoothly with the input sequence $x$, with a Lipschitz constant that does not explode with $T$. This contrasts with chaotic systems where a small change in initial conditions or inputs can lead to exponentially diverging states over time.

# E. Additional Experiments and Details

### E.1. Quantitative Generalization: Base-2 Evaluation

While the previous examples (Parity, Addition) demonstrated the ability to track discrete states, they did not test the capacity for *quantitative* accumulation over unbounded domains. We consider the task of *Base-2 Integer Evaluation*: mapping a binary string $x \in \{0,1\}^L$ to its integer value $f(x) = \sum_{t=1}^{L} x_t \cdot 2^{L-t}$. This requires the hidden state to grow exponentially with sequence length ($v_t = 2v_{t-1} + x_t$), serving as a stress test for the "Linearity vs. Saturation" hypothesis.

**Experimental Setup.** We trained models on sequences of length $L = 64$. To isolate architectural limitations from hardware precision limits, all operations were performed in *Double Precision (Float64)*. We compared the Rational Transductor against a Transformer and an LSTM.

**Results.** As shown in Figure 6b, this task reveals a sharp expressivity gap. Both the LSTM and Transformer fail completely, converging to the variance of the dataset ($MSE \approx 8.4 \times 10^{-2}$). For the LSTM, the gradient signal vanishes through 64 layers of saturating non-linearities ($\tanh$); for the Transformer, the attention mechanism cannot resolve positional weights spanning 19 orders of magnitude ($2^{64} \approx 1.8 \times 10^{19}$) amidst softmax noise. In contrast, the Rational Transductor exploits its linear recurrence to propagate gradients without attenuation, learning the exact Horner scheme to machine precision ($MSE \approx 5.9 \times 10^{-9}$).

### E.2. Hyperparameter Specifications

To ensure reproducibility, we detail the exact hyperparameters used for the experiments in Section 6. All models were implemented in PyTorch and trained on a single NVIDIA T4 or A100 GPU. Optimization was performed using AdamW (Loshchilov & Hutter, 2019) or Adam (Kingma & Ba, 2015).

Table 2 summarizes the configurations for all four synthetic tasks. Note that for the *Long-Integer Addition* task, we used a curriculum learning strategy where the training sequence length was sampled uniformly from $U[10, 40]$ at each step to encourage robust generalization. LSTM baseline uses standard PyTorch initialization.

*Table 2.* Hyperparameters for Rational Transductor Experiments. (RT: Rational Transductor, TF: Transformer.)

| Config / Task | Modulo Counting (Section 6.1) | Length Gen. (Section 6.2) | Long Addition (Section 6.4) | Base-2 Eval (Appendix E.1) |
|---|---|---|---|---|
| *Model Architecture* | | | | |
| Hidden Dim ($d_{\text{model}}$) | 32 | 32 | 32 | 12 (RT) / 32 (TF) |
| Rational State Dim ($d_{\text{rat}}$) | 8 | 8 | 4 | 12 |
| Layers | 2 | 2 | 2 | 1 (RT) / 3 (TF) |
| Heads | 4 | 4 | 4 | 4 |
| Parametrization | Orthogonal (Cayley) | Orthogonal (Cayley) | Stochastic (Softmax) | Affine (General) |
| *Optimization* | | | | |
| Seq Length ($L_{\text{train}}$) | 50 | 40 | $U[10, 40]$ | 64 |
| Batch Size | 64 | 64 | 64 | 32 |
| Optimizer | AdamW | AdamW | AdamW | Adam |
| Learning Rate | $5 \times 10^{-3}$ | $5 \times 10^{-3}$ | $5 \times 10^{-3}$ | $1 \times 10^{-2}$ |
| Scheduler | None | Cosine Annealing | None | Cosine Annealing |
| Gradient Clip | 1.0 | 1.0 | 1.0 | 1.0 |
| Training Steps/Epochs | 3,000 Steps | 3,000 Steps | 4,000 Steps | 60 Epochs |
| Loss Function | Cross Entropy | Cross Entropy | Cross Entropy | MSE |
| Precision | Float32 | Float32 | Float32 | Float64 |

### E.3. Near-Identity Initialization

To stabilize training, we initialize the rational head to act as a near-perfect integrator. Regardless of the chosen parameterization (Cayley, DPLR, or Affine), we initialize parameters such that the transition matrices start close to the identity ($M_\sigma \approx I$).

Specifically:

- **Diagonal/Rotation Terms:** For Cayley parameterization, the skew-symmetric matrix $A_t$ is initialized near zero (implying $M \approx I$). For DPLR, the diagonal term $D_\sigma$ is initialized to 1 (or sampled from $U[1 - \epsilon, 1]$ with $\epsilon \ll 1$).

- **Perturbation Terms:** Any low-rank update terms (e.g., $U_\sigma, V_\sigma$) are initialized independently from $\mathcal{N}(0, \nu^2)$ with sufficiently small variance ($\nu^2 \approx 0$).

This initialization bias ensures that at step 0, the model acts as a near-perfect integrator capable of preserving state information over long horizons. The optimizer then gradually learns to "forget" irrelevant information by deviating from the identity, rather than struggling to learn "remembering" from a chaotic or vanishing initialization.

### E.4. Statistical Significance and Stability

**Stability of Initialization:** The *Near-Identity Initialization* described in Section E.3 provides a highly stable starting point for learning algebraic structures. In our experiments (specifically Subsection 6.1 and Subsection 6.2), the Rational Transductor converged to 100% training accuracy within the first 500-1000 steps in every trial. The loss curves exhibit a deterministic drop characteristic of solving a convex-like problem in the lifted state space, rather than the high-variance *grokking* often seen in standard Transformers on these tasks.

**Variance:** We repeated the *Modulo Counting* and *Length Generalization* experiments across $N = 5$ random seeds.

- **Rational Transductor:** Achieved 100% accuracy on the training distribution in 5/5 runs. Generalization to length $L = 500$ was 100% in 5/5 runs. Standard deviation $\sigma < 0.01\%$.

- **Baseline Transformer:** Consistently failed to generalize beyond $L_{\text{train}}$, with accuracy collapsing to the random baseline (20%) in all seeds, exhibiting negligible variance in its failure mode.

We conclude that the reported performance gap is due to the fundamental difference in inductive bias (Recurrent vs. Attention), not initialization luck.

### E.5. Additional Notes on Experiments

For the Computational Efficiency experiment, we note that recent methods like ParaRNN (Danieli et al., 2025) enable quasi-parallel training of non-linear RNNs via iterative linearization (Newton's method), achieving theoretical $O(k \log T)$ depth. However, this approach requires multiple forward scans per update step. We benchmark against the standard exact sequential implementation used in most production baselines.

For the Quantitative Generalization experiment, To strictly isolate architectural limitations from hardware precision limits, all operations were performed in *Double Precision (Float64)*. To fit within the dynamic range of double precision floating point arithmetic, target integer values were normalized to the unit interval $[0, 1]$ via scaling by $2^{-L}$. We note that for sequence lengths $L > 53$, the integer values $2^L$ exceed the 53-bit significand precision of IEEE 754 Double Precision (Float64). The reported "machine precision" MSE in Figure 6b reflects the limit of representable hardware floating point accuracy for this sequence length, rather than a theoretical failure of the model.

## F. Limitations

The expressivity of the rational head is bounded by its state dimension $d$, which corresponds to the rank of the underlying Hankel matrix, the *Algebraic Capacity* of the model. Recognizing regular languages whose minimal DFA has $n$ states requires $d \geq n$, and the parallel scan cost scales as $O(d^2 T)$. Consequently, very large-state automata may demand either large $d$ (increasing compute) or architectural innovations (e.g., the stacked cascading of Section B.2). Exploring this capacity-complexity trade-off in practical settings is an important direction for future work.

