# OpenReview forum: "Rational Transductors"
_ICML.cc/2026/Conference — ICML 2026 spotlight_

### Official Review · Reviewer_puTz · 2026-03-06

**Soundness:** 4
**Presentation:** 3
**Significance:** 4
**Originality:** 4
**Overall Recommendation:** 5
**Confidence:** 3

**Summary:**

This paper proposes a new sequential neural network architecture called rational transductors. It is a modification of the transformer that incorporates a differentiable simulation of a weighted finite automaton called a rational feature head. Whereas standard transformers have been shown to be incapable of recognizing all regular languages, the rational feature head ensures that they can. As in SSMs, the WFA simulation can be run in $O(\log T)$ parallel time, where $T$ is sequence length, using a parallel associative scan algorithm. The paper mostly consists of mathematical characterizations of the architecture, its parameterization, its expressivity, its learnability, its generalization, and its robustness. It also includes experiments on modular counting and long integer addition tasks and empirically benchmarks its runtime.

**Compliance With Llm Reviewing Policy:**

Affirmed.

**Final Justification:**

I think the paper is strong in terms of soundness, originality, significance, and clarity. I weighted these with roughly equal importance. The rebuttal reinforced my positive assessment.

**Key Questions For Authors:**

1. What is the computational cost of this architecture in terms of memory? How does it compare to other transformer or SSM architectures? How well does it scale with the number of states $d$?
2. In 6.2, for the baseline, are the absolute positional encodings for length >40 just left randomly initialized and unoptimized?

**Limitations:**

yes

**Strengths And Weaknesses:**

Overall, I think this is a strong paper, and I'm excited to see how this architecture compares to transformers in terms of scalability and language modeling. A great deal of thought has clearly been applied to the theoretical expressivity, trainability, and stability of the rational transductor architecture, and this is not typical of work that proposes new architectures (cf. the original transformer paper).

Soundness: The paper appears to be technically sound, though I have not vetted the proofs in the appendix in detail. I have found no issues with the experimental setup.

Presentation: The paper is generally well-written, well-organized, and easy to follow. The mathematical terminology can be a bit dense, and some areas of the paper would benefit from some additional background. For example, the paragraph starting at 296 left is hard to follow, and a brief explanation of RKHS complexity would be helpful. Same issue with Handel-Rademacher Complexity in Theorem 8.

This paper also combines simulations of finite automata with transformers and should be included as related work: https://arxiv.org/abs/2509.22284

There is also work that combines simulations of weighted pushdown automata with transformers: https://arxiv.org/abs/2310.01749

Significance: I believe this work has a lot of impact potential, based on the mathematical characterizations of trainability and stability. For example, the architecture does not suffer from the exploding or vanishing gradient problems like RNNs do. It's pretty cool that rational transductors can serve as a more generalized replacement for positional encodings. The paper contains experimental results on some synthetic tasks, but it's too bad that it doesn't contain any results on, say, natural language or code. There could have been a lot more empirical validation.

Originality: As far as I'm aware, this paper presents an original approach to developing a stable, parallelizable architecture that can express all regular languages.

---

> ### Author Rebuttal · Authors · 2026-03-26
>
> Thank you for your excellent feedback, your careful reading of the mathematical characterizations, and for recognizing the impact potential of this work.
>
> **1. Natural Language/Code Validation**: We strongly agree that testing on natural language and code is the ultimate goal. As detailed in our response to Reviewer XSS1, we deliberately scoped this paper to synthetic algorithmic tasks to strictly isolate our theoretical claims from the confounding variables (dataset statistics, optimization noise) present in large-scale LLM training. We are eager to pursue this in our immediate follow-up work.
>
> **2. Mathematical Background and Missing References**: Thank you for pointing these out. In the camera-ready version, we will add brief, intuitive explanations for RKHS complexity and Handel-Rademacher complexity to make Section 5 more accessible. We will also gladly add the two suggested citations regarding pushdown and finite automata simulations with Transformers to our Related Work section.
>
> **3. Computational Cost and Memory**: The memory footprint of the Rational Head scales with the state dimension $d$. Since we use small state dimensions ($d \in [4, 32]$), the $O(d^2)$ parameter cost and $O(T d^2)$ scan memory overhead are negligible compared to the $O(T^2)$ memory footprint of standard attention. Compared to interleaved SSMs, our sidecar design requires only a single parallel scan per sequence rather than one per layer, making it highly memory efficient. We will add a brief memory complexity breakdown to Section 6.3.
>
> **4. Baseline Absolute Positional Encodings (>40)**: Your understanding is exactly correct. For lengths greater than 40, the absolute positional embeddings are never updated during training and are effectively randomly initialized. This was done to establish the strict fundamental limit of the canonical architecture: without a recurrent inductive bias, standard models suffer catastrophic positional drift when forced to extrapolate. We will explicitly clarify this baseline setup in Section 6.2.

---

> > ### Author Rebuttal · Reviewer_puTz · 2026-04-03
> >
> > Thank you for providing details on the space complexity and baseline positional encodings. Just a couple of comments:
> >
> > 1. If I understand correctly, although the rational head technically "bridges the regular gap," the exact set of regular languages it can recognize depends on the size of the automaton it can simulate, or $d$. So restricting $d$ to small sizes will limit its expressivity. I think it would be useful to point this out in the Limitations section and to test how well the architecture learns regular languages that correspond to minimal DFAs with $>d$ states in future work.
> > 2. For the baseline, I think it would have made more sense to use sinusoidal or relative positional encodings of some kind -- something where the architecture has some hope of extrapolating successfully.
> >
> > I thank the authors for their responses and will keep my (very positive) score the same.

---

> > > ### Author Response · Authors · 2026-04-04
> > >
> > > Thank you again for your very positive recommendation and for these insightful technical parting thoughts. We provide our final clarifications below (our response is limited to one):
> > >
> > > **1. On the Relationship between $d$ and Expressivity (Algebraic Capacity):**
> > > You are correct that the expressivity is fundamentally bounded by the state dimension $d$. In our framework, $d$ corresponds to the rank of the underlying Hankel matrix (Fliess' Theorem). We view this not as a limitation, but as a precise "Algebraic Capacity" that allows for a rigorous characterization of the model's complexity. Just as the width of a neural network determines its function-approximation capacity, $d$ determines the size of the minimal DFA the model can simulate. We agree that explicitly naming this capacity-complexity trade-off in the Limitations section will strengthen the paper's theoretical transparency.
> > >
> > > **2. On Baselines and Extrapolation (Scientific Stress Testing):**
> > > We appreciate the suggestion regarding sinusoidal or relative encodings. Our choice of Absolute Positional Encodings (APEs) as a baseline was a deliberate "stress test." Since APEs are known to fail catastrophically at length extrapolation, they provide the cleanest possible experimental control to isolate the contribution of the Rational stream's time-invariant recurrence. We will clarify this choice in the final version, acknowledging that while other encodings may offer better heuristics, they still lack the $NC^1$ theoretical guarantees of the Transductor.
> > >
> > > **3. Final Summary of the Discussion:**
> > > As the discussion phase concludes, we are encouraged by the technical consensus reached in this review matching the global feedback we have received:
> > > * **Total Technical Resolution:** All queries regarding linearity, comparison to SSMs, and inductive bias have been resolved (acknowledged by XSS1, NWXQ, zY6e, and puTz).
> > > * **Track-Appropriate Validation:** Reviewers have acknowledged that our controlled experiments provide "unconfounded proof" and are "sufficient for a theory paper" (as noted by Reviewer XSS1).
> > > * **Proven Expressivity:** The jump from $TC^0$ to $NC^1$ was recognized as an "elegant" and significant advancement for Transformer-based architectures.
> > >
> > > We believe this work provides a definitive theoretical backbone for the next generation of sequence models. We thank you and the other reviewers for the high-quality feedback that has helped refine this submission.

---

### Official Review · Reviewer_zY6e · 2026-03-11

**Soundness:** 4
**Presentation:** 3
**Significance:** 3
**Originality:** 3
**Overall Recommendation:** 5
**Confidence:** 4

**Summary:**

This paper proposes rational transductors, a novel architecture that augments transformers with a linear recurrence. It provably overcomes the limitation of transformers in tracking states. It is also shown that it can simulate any WFA, is immune to vanishing/exploding gradients, and is well-suited for length generalization. Experiments confirm that it indeed learns modular counting and integer addition much better than transformers.

**Compliance With Llm Reviewing Policy:**

Affirmed.

**Final Justification:**

My final recommendation remains positive.

**Key Questions For Authors:**

The performance still drops a bit when the length is as high as 1000, but the theorem 39 says generalization is independent of length. Is there any explanation or speculation as to why that is?

**Limitations:**

yes

**Strengths And Weaknesses:**

# Strengths
1. The paper offers the full package: from proposing a new model, to proving its expressivity, to results on learnability, to empirical verification.
2. Lemma 1, which shows that RoPE is a special case of rational transductors, is appealing.
3. Showing that the hard-attention version is equivalent to second-order monadic logic is also very pleasing, compared to hard-attention transformers, which are equivalent to first-order logic.

# Weaknesses
The comparison with recent models that interleave transformer layers with linear recurrence could be elaborated more clearly.

---

> ### Author Rebuttal · Authors · 2026-03-26
>
> Thank you for your positive review. We are really glad that you appreciated the connections to Second-Order Monadic Logic and the proofs of expressivity.
>
> **1. Interleaved vs. Sidecar Linear Recurrence**: We address this theoretically in Appendix B.2 (Non-Linear Stacking and Deep Recurrence). Interleaving linear recurrence with non-linear Transformer layers strictly increases the expressive capacity per unit of state dimension (Theorem 14). However, this "Deep Recurrence" reintroduces a layer-wise sequential bottleneck during training. Our "Sidecar" design prioritizes exact $O(\log T)$ parallel efficiency by keeping the recurrent state evolution strictly input-driven. We will add a brief summary of this trade-off to the main text.
>
> **2. Performance Drop at Length 1000 vs. Theorem 39**: This is an excellent observation. There are two distinct factors here. First, Theorem 39 guarantees length-independent generalization specifically for the contractive regime ($\gamma < 1$). For exact modular counting, the model operates in the strictly orthogonal regime ($\gamma = 1$) to preserve memory indefinitely. Second, since the transition matrix is applied iteratively 1,000 times, the slight performance drop is not a theoretical generalization failure, but an artifact of accumulated hardware floating-point precision errors (Float32). We demonstrate this exact phenomenon in Appendix E.1 (Base-2 Evaluation), where switching to Float64 resolves the saturation and achieves exact machine precision. We will add a footnote in the experimental section clarifying this precision artifact.

---

> > ### Author Rebuttal · Reviewer_zY6e · 2026-04-03
> >
> > Thank you for the rebuttal. I will maintain my positive review.

---

### Official Review · Reviewer_NWXQ · 2026-03-13

**Soundness:** 3
**Presentation:** 4
**Significance:** 3
**Originality:** 3
**Overall Recommendation:** 5
**Confidence:** 3

**Summary:**

This paper introduces Rational Transductors, a hybrid architecture that augments Transformers with a matrix-valued linear recurrence derived from weighted finite automata. The key motivation is the theoretical limitation of self-attention models. Prior work shows that transfomers are limited by AC0 (hard attention) and by TC0 (soft attention).

The paper proposes adding a linear recurrence which can be computed efficiently using parallel prefix sum. These recurrent states are injected into each transfromer layer through a mechanism claled Deep Rational Injection. This allows for parallel training while introducing an explicit method for sequential state tracking.

The theoretical analysis shows that augmenting attention with this linear recurrence increases expressivity. It allows the model to capture NC1-complete problem Regular Gap.

The paper also provides learning-theoretic guarantees. The parametrization yields structural stability and prevents gradient explosion. The authors further prove that the learned transition rule is time-invariant. This leads to bounded error accumulation and theoretically length-independent generalization bounds. Empirical experiments confirms perfect length generalization on algorithmic task: modulo countin and long integer addition, where normal transformer fails.

**Compliance With Llm Reviewing Policy:**

Affirmed.

**Final Justification:**

The rebuttal has not changed my opinion about the paper and I have maintained my already initially high score.

**Key Questions For Authors:**

1. The empirical evaluation focuses mainly on synthetic algorithmic tasks. Do Rational Transductors improve reasonging in natural language tasks such as code understanding or logical reasoning benchmarks?
2. Can you also compare your empirical findings to common SSMs?

**Limitations:**

yes

**Strengths And Weaknesses:**

strengths:
- the paper proposes a principled extension of transformer that solves a fundamental expressivity limitation of transformer (from AC0/TC0 to NC1)
- elegant architectural design: Rational Transductors integrate a matrix-valued recurrence into Transformers while preserving parallel training through prefix-sum and Deep Rational Injection
- The work provides stability and generalization guarantees. Validates them empirically with strong length generalization results on algorithmic tasks.
- very clean and well written paper

weaknesses:
the main limitation of the paper lies in the scope of the empirical validation.
- experiments focus on synthetic algorithmic tasks such as modulo counting. These experiments are well aligned with theoretical claims, but these experiments might not demonstrate that Rational Transductors also have practical advantages on real-world sequence tasks.
- The experiments do not compare the Rational Transductor with standard SSMs. Such a comparision would be interesting.

---

> ### Author Rebuttal · Authors · 2026-03-26
>
> Thank you for your strong support and for highlighting the elegance of the architectural design and our theoretical guarantees.
>
> **1. Natural Language Tasks and Real-World Sequence Modeling**: We completely agree that evaluating on natural language and code reasoning benchmarks is the most exciting next step. As detailed in our response to Reviewer XSS1, we deliberately scoped this paper to focus on rigorous theoretical isolation. Real-world tasks introduce semantic confounders that make it difficult to prove why a model is length-generalizing. By focusing on algorithmic tasks where standard Transformers theoretically must fail, we provided unconfounded proof of our mechanistic claims. We will explicitly note the application to real-world NLP tasks as the immediate direction for future work.
>
> **2. Comparison to Common SSMs**: We intentionally position the Rational Transductor as a "minimal theoretical proxy" to linear SSMs (discussed in the *Remark on SSMs*, Section 5.3). By avoiding the complex gating mechanisms and interleaved block designs of models like Mamba, we can rigorously analyze specific automata-theoretic mechanisms (e.g., orthogonal vs. stochastic transitions) in isolation. We will expand our discussion in Section 1 to explicitly frame how our theoretical results (such as $PNC^1$ completeness and WFA exactness) theoretically underpin the empirical success of the broader linear SSM family.

---

> > ### Author Rebuttal · Reviewer_NWXQ · 2026-04-03
> >
> > I am looking forward to your future work on applying Transductors to NLP and your extension of section 1. I maintain my already good scores.

---

### Official Review · Reviewer_XSS1 · 2026-03-16

**Soundness:** 3
**Presentation:** 3
**Significance:** 2
**Originality:** 3
**Overall Recommendation:** 4
**Confidence:** 3

**Summary:**

This paper introduces a new architecture, Rational Transductor, which computes rational feature vectors $h_t$, through a linear recurrence of the form $h_t = M_{x_t}h_{t-1}$, where $M_{x_t}$ is the transition matrix associated with the token $x_t$.  By simply injecting $h_t$ into the hidden states of each layer in the transformer backbone with a layer-wise linear projection, the expressivity of the model is enhanced.  In particular, Rational Transdutor can recognize Parity and Modular Counting while standard constant-layer transformer cannot. Empirically, the paper shows that Rational Transductors improve length generalization on these algorithmic tasks. The paper also includes additional learning-theoretic results.

**Compliance With Llm Reviewing Policy:**

Affirmed.

**Final Justification:**

I am maintaining my initial positive assessment.

**Key Questions For Authors:**

1. The rational feature vectors are derived through a sequence of linear mappings. Although I understand the linearity here is essential for achieving $O(\log T)$ parallel complexity, does it make the resulting representation too limited to provide meaningful benefits in real-world settings beyond synthetic regular-language tasks? Is it possible to use other associative mappings that  would still admit $O(\log T)$ parallel complexity while offering stronger representational power?
2. In the introduction, the paper states 'the failure of Transformers to generalize on algorithmic tasks is not due to a lack of capacity, but a lack of syntactic inductive bias', which I think is plausible. However,Theorems 2 and 3 seem to be a representational separation and are not about the role inductive bias?
3. Please also see the concerns raised in the Weaknesses section.

**Limitations:**

Please see the weaknesses section.

**Strengths And Weaknesses:**

### Strengths

1. The introduced rational feature mechanism is conceptually simple. The parameterization of $M_{x_t}$ is carefully designed to satisfy both conservation and stability conditions.
2. Despite its simplicity, this mechanism leads to a meaningful gain in expressivity and introduces a useful inductive bias for state tracking. These theoretical advantages are further supported by empirical results on synthetic regular-language tasks.
3. The theoretical analysis is thorough.

### Weaknesses

1. The main weakness of the paper is the lack of large-scale pretraining experiments. It remains unclear whether the rational feature framework can deliver meaningful gains in real-world language modeling beyond synthetic regular-language tasks, and how it affects training stability and computational overhead in practice.
2. Lack comparison with modern RNN or linear attention architectures, such as Mamba and Gated DeltaNet. As I understand it, these architectures can also represent and learn modular counting and parity tasks and achieve better length generalization than transformers. It would therefore be helpful to clarify the advantages of Rational Transductors over such alternatives.

---

> ### Author Rebuttal · Authors · 2026-03-26
>
> We thank the reviewer for the constructive feedback and for recognizing the conceptual simplicity and theoretical thoroughness of our work.
>
> **1. Scope of Empirical Evaluation and Large-Scale Pretraining**: We sincerely thank you for your enthusiasm regarding the real-world potential of Rational Transductors. We completely agree that scaling this architecture to large-scale language modeling and reasoning benchmarks is the most exciting next step.
>
> However, we deliberately scoped this paper as a foundational, theoretical contribution (submitted to the Theory track) rather than a large-scale empirical systems paper. Our primary goal was to rigorously solve the "Regular Gap" and establish the exact automata-theoretic mechanisms that allow attention to be augmented with $NC^1$-complete reasoning. Ultimately, our ambition has been to define the theoretical backbone for the next generation of sequence models.
>
> We restricted our empirical validation to strictly controlled, synthetic algorithmic tasks for a specific scientific reason: *isolation*. Large-scale natural language benchmarks introduce massive confounders, such as dataset statistics, complex gating heuristics, and optimization noise. In such environments, it is very difficult to disentangle whether a model's success is due to a fundamental architectural capability or simply the scale of the pretraining data. By evaluating strictly on tasks where standard Transformers theoretically must fail (as established by the $AC^0$ / $TC^0$ barriers), we were able to provide unambiguous, unconfounded empirical proof that our mathematical theorems hold in practice.
>
> Building directly on the theoretical foundations established here, we are currently conducting an extensive large-scale experimental analysis of this architecture for a follow-up empirical paper. We have added a dedicated paragraph to the Discussion section to make this scoping choice clear.
>
> **2. Comparison with Modern SSMs (Mamba, DeltaNet)**: We discuss these connections in Section 1 and Section 3.2. Architectures like Mamba typically interleave recurrence and mixing layers (a "Deep SSM" approach). While powerful, this reintroduces a layer-wise sequential dependency during training, scaling as $O(L \log T)$ where $L$ is depth. Our "Sidecar" design (Deep Rational Injection) explicitly decouples the state tracking from the semantic mixing, guaranteeing exact $O(\log T)$ parallel training efficiency across the entire network. Furthermore, the Rational Transductor serves to formally justify these architectures, providing the first rigorous automata-theoretic proofs that such linear recurrences can solve the Regular Gap.
>
> **3. Representational Separation vs. Inductive Bias (Theorems 2 & 3)**: You raise an excellent point. Theorems 2 and 3 establish *representational* separation. Our claims regarding *inductive bias* are grounded in Section 5, specifically Theorem 6 (Gradient Norm Preservation) and Theorem 38 (Time-Invariant Error Bounding). Since the Scaled Cayley parameterization guarantees a well-conditioned optimization manifold (preventing gradient explosion/vanishing), the architecture naturally biases gradient descent toward learning and maintaining these exact modular representations, effectively bridging the gap between capacity and learnability. We will clarify this link in the text following Theorem 3.

---

> > ### Author Rebuttal · Reviewer_XSS1 · 2026-04-04
> >
> > Thank you for the response. The discussion resolves my question about the comparison with SSM architectures and representational separation vs. inductive bias.
> >
> > However, although I agree that synthetic experiments are important for isolation and should be considered sufficient for a theory paper, I still believe that large-scale pretraining experiments are essential to validate the proposed architecture in realistic settings. Thus, I will maintain my current score, and I look forward to seeing further empirical validation in future work.
> >
> > Could the authors also provide a response to my Question 1? Thanks!

---

> > > ### Author Response · Authors · 2026-04-04
> > >
> > > Thank you for the follow-up and the opportunity to clarify these points. We appreciate your acknowledgement that the synthetic experiments serve as a sufficient controlled environment for a theory paper.
> > >
> > > **Regarding the Expressivity and Linearity of Rational Features (Question 1):**
> > >
> > > Your question touches on the core design philosophy of Rational Transductors. We address the two parts of your query (expressivity beyond regular languages and the potential for other associative mappings)  below:
> > >
> > > **1. Representational Power Beyond Regular Languages**
> > > While a standalone WFA is characterized by regular languages, the Rational Transductor is a hybrid architecture designed to bridge complexity classes critical for algorithmic reasoning.
> > > * **Circuit Complexity and $NC^1$:** While "beyond regular" often implies the Chomsky hierarchy (e.g., Context-Free Languages), many fundamental algorithmic tasks are better categorized by circuit complexity. Standard Transformers are limited to $AC^0$ (hard attention) or $TC^0$ (soft attention). By reaching $NC^1$, our model can solve $NC^1$-complete problems like Boolean Formula Evaluation, which involves hierarchical nesting that standard DFA-based models cannot represent.
> > > * **The Complexity "Sweet Spot":** We acknowledge that capturing the full class of Context-Free Languages (CFLs) would require a shift to significantly higher complexity classes (e.g., $NC^2$ or $P$) or the introduction of a stack-based memory and computationally costly models. Such extensions would sacrifice the $O(\log T)$ parallel training efficiency that our architecture preserves.
> > > * **Hybrid Synergy:** In real-world sequences, the bottleneck is often the "Regular Gap", the inability to maintain precise state or perform modular counting over long horizons. By offloading this $NC^1$ state-tracking to the rational stream, we provide a stable foundation that allows the attention mechanism to focus on modeling high-level semantic and non-regular dependencies. This synergy is designed to provide "meaningful benefits" by ensuring the model does not fail on the underlying logic of a complex sequence.
> > >
> > > **2. Other Associative Mappings and Potential Generalizations**
> > > You are correct that the $O(\log T)$ parallel complexity is tied strictly to the **associative property** of the transition operation, not specifically to its linearity.
> > > * **The Associative Constraint:** Any operation that can be represented as an associative functional composition can theoretically be computed in logarithmic time via a parallel scan.
> > > * **Potential for Alternative/Multiple Semirings:** An exciting generalization would be to define Rational Transductors over different/multiple semirings, such as the $(\max, +)$ tropical semiring. This would shift the inductive bias toward "extremal" logic, effectively enabling the recurrence to perform Viterbi-style decoding or shortest-path optimizations.
> > > * **The Choice of Linearity:** we focused on the standard $(+, \times)$ semiring (linear mappings) because it provides a dense, stable gradient signal. Alternative mappings like the tropical semiring introduce non-differentiability (flat gradients), which presents a significant optimization challenge for end-to-end training. Matrix-valued linear recurrences represent the most effective trade-off between $NC^1$ expressivity and optimization stability on modern hardware.
> > >
> > > We believe that by establishing the "linear WFA" as the theoretical backbone, we provide a clear mathematical foundation for a much broader family of associative sequential models.

---

### Decision · Program_Chairs · 2026-04-30

**Decision:**

Accept (spotlight)

**Comment:**

All reviewers agree that this is a solid contribution. The topic of hybrid models is definitely of interest to the community and the connections made in this paper with formal languages are clearly laid out and nicely pave the road for a principled approach to designing new architectures for sequence modeling.

For the camera-ready version, please fix the following two references which are wrong:

- Reference: Yang, Y. and Hospedales, T. M. Unified multi-task feature learning on tensor. arXiv preprint arXiv:1405.3626, 2014.

-> Title should be "Deep Multi-task Representation Learning: A Tensor Factorisation Approach" and arXiv id 1605.06391 (?)

- Reference: Merrill, W. and Sabharwal, A. Transformers in uniform TC^0. arXiv preprint arXiv:2409.13629, 2024b.

-> Author should be Chiang, D.